# On the value of reanalyses prior to 1979 for dynamical studies of stratosphere-troposphere coupling

Peter Hitchcock[1]

[1]Laboratoire de Météorologie Dynamique, École polytechnique, Palaiseau, France

**Correspondence:** Peter Hitchcock (peter.hitchcock@lmd.polytechnique.edu)

**Abstract.** Studies of stratosphere-troposphere coupling, particularly those seeking to understand the dynamical processes underlying the coupling following extreme events such as major stratospheric warmings, suffer significantly from the relatively small number of such events in the 'satellite' era (1979 to present). This limited sampling of a highly variable dynamical system means that composite averages tend to have large uncertainties. Including years during which radiosonde observations of the stratosphere were of sufficiently high quality substantially extends this record, reducing this sampling uncertainty by up to 20%. Moreover, many open questions in this field involve aspects of tropospheric dynamics likely to be better constrained by 'conventional' (i.e. radiosonde and surface-based) observations.

Based on an inter-comparison of reanalyses, a quantitative case is made that for many purposes the improved sampling obtained by including this period outweighs the reduced precision of the reanalyses in the Northern Hemisphere. Studies of stratosphere-troposphere coupling should therefore consider the use of this period when using reanalysis data. These results also support continued attention on this period from centres producing reanalyses.

## 1 Introduction

One of the central challenges to the detailed study of the large-scale coupling between the stratosphere and the troposphere is the relatively limited record of high quality, global observations. In the absence of more insightful modes of analysis, quantifying the dynamical processes relevant for the coupling requires large samples to isolate them from unrelated dynamical variability. Despite the availability of nearly four decades of global satellite-based observations, the length of the observational record remains a fundamental limitation to this statistical approach. This is demonstrated explicitly here, as well as by another closely related contribution (Gerber and Martineau, 2018) to the SPARC Reanalysis Intercomparison Program (SRIP; Fujiwara et al., 2017).

The coupling between the stratosphere and the troposphere remains a significant source of uncertainty in projected climate changes over the coming century (Manzini et al., 2014; Simpson et al., 2018), as well as an important source of skill in seasonal forecasting (Sigmond et al., 2013). Global models exhibit a diversity of stratospheric circulation (Manzini et al., 2014) and variability (Charlton-Perez et al., 2013; Taguchi, 2017), and of tropospheric responses to stratospheric variability (Hitchcock and Simpson, 2014). Observations of the true circulation can be used to identify which models are correctly representing these processes, but this relies on comparing the time-averaged behaviour of the models to the observations, and the large

interannual variability in the observed circulation means that the sampling uncertainty remains large. Accounting for sampling error in such large-scale dynamical phenomena is a major concern for many other dynamical questions, including identifying regional signals of climate change and teleconnection patterns (e.g., Deser et al., 2017).

Studies of observed stratosphere troposphere coupling often rely on reanalysis products, which combine a wide range of observations with global forecast models (see Fujiwara et al., 2017, for a comprehensive discussion). Two of the older products, ERA-40 and NCEP-NCAR R1, begin in 1957 and 1948, respectively, dates which coincide with significant extensions of the global radiosonde observing network. Many more recent products (ERA-Interim, MERRA, MERRA-2, CFSR) by contrast cover only the period from 1979 onwards, after the availablility of sounding data from Microwave Sounding Unit (MSU) and Stratospheric Sounding Unit (SSU) instruments. It is convenient to label the period after 1979 the 'satellite' era, though it is worth noting that a number of satellite data products exist prior to 1979, as discussed by Uppala et al. (2005). Amongst the more modern products only JRA-55 begins prior to the satellite era, in 1958. However, both ERA-5 and JRA-3Q, two newer products unavailable at the time of writing, are expected to cover the pre-satellite era as well.

For the purposes of the present work, the 'radiosonde' era will refer to the period from 1958 to 1978, although radiosonde data exists prior to this period and continues to be important afterwards. There is no general consensus amongst studies of stratosphere-troposphere coupling as to whether to include the radiosonde era. This is complicated by the fact that the coverage of ERA-40 ends in 2002, leaving out the most recent (and best-observed) decade and a half. Some studies have made use of the older reanalysis products ERA-40 and NCEP-NCAR R1 alone (Charlton and Polvani, 2007; Mitchell et al., 2013) while others consider exclusively the satellite record (Dunn-Sigouin and Shaw, 2014; Kodera et al., 2015; Birner and Albers, 2017). Still others choose to merge multiple reanalyses, using an older product for the radiosonde era and a more modern product for the satellite era (Hitchcock et al., 2013; Lehtonen and Karpechko, 2016). The value of JRA-55 as a single modern product that spans both the radiosonde and satellite eras is thus evident, (and as such it will be privileged in the analysis that follows) but the question remains whether the observational record during the radiosonde era is of 'sufficiently' high quality to be worth considering.

The first identification of a stratospheric sudden warming is credited to Scherhag (1952) and much was known about their dynamics prior to the availability of a long satellite-based observational record (e.g., Matsuno, 1971; Labitzke, 1977; McIntyre, 1982), largely on the basis of radiosonde observations. Moreover, a successful five-day forecast of the sudden warming that occurred in January 1958 initialized from ERA-40 has been demonstrated (Simmons et al., 2005). All of this suggests that the observational record prior to 1979 is of real value in constraining the behaviour of the coupled stratosphere-troposphere system around stratospheric sudden warmings.

The immediate goal of this work is to evaluate the representation of a number of quantities of interest to the problem of stratosphere-troposphere coupling in the radiosonde era, in view of coming to a more quantitative assessment of their value. For the Northern Hemisphere the arguments given below clearly indicate their value. However, since this judgement depends on the specific quantity of interest, a broader goal is to discuss how to answer this question more generally. Indeed, the same arguments should apply to the study of many other features of the large-scale atmospheric circulation, particularly of those

phenomena with large spatial scales and characteristic timescales of the order of weeks to months. The same approach could also be applied in principle to the period prior to 1958, although no effort has been made to do so here.

This evaluation is based on the availability of multiple reanalysis products. Since in general the different reanalyses assimilate subsets of the same observational record into distinct forecast models, the level of agreement provides a simple measure of how strongly the observations constrain the quantity in question. This method has caveats in that the underlying forecast models may share biases that result in them getting consistently wrong answers. More critically, the availability of only one modern reanalysis product that covers the radiosonde era (and assimilates radiosonde data) means that this comparison must be based in part on older reanalyses with known deficiencies (e.g., Long et al., 2017). Nonetheless, as will be argued below, the agreement is close enough in the Northern Hemisphere to suggest that this period has real value for carrying out many classes of dynamical studies. This is broadly consistent with the conclusions of Gerber and Martineau (2018) and of Hersbach et al. (2017), which explicitly examined the value of upper-air observations over the period 1939 to 1967 in an experimental reanalysis product.

The outline of this paper is as follows. The reanalysis data considered here is described in Section 2. Section 3 presents, as an initial example, a discussion of the time series of zonal mean zonal wind at 10 hPa and 60° N that is central to the identification of major sudden stratospheric warmings. Section 4 presents more general criteria for determining when the radiosonde era should be included. These criteria are then discussed in Section 5 as they apply to wider variety of zonal mean quantities, including fluxes of heat and momentum that are relevant to stratosphere-troposphere coupling. Section 6 presents conclusions and a discussion.

## 2 Reanalysis data

Zonally averaged output from the 12 reanalysis products listed in Table 1 are considered here. Of these reanalyses, five (JRA-55, NCEP-NCAR, ERA-40, 20CR v2, and ERA-20C) include the period from 1958 through 1978. Two reanalysis products (20CR v2 and ERA-20C) extend further back but do not assimilate upper-air observations; following the nomenclature of Fujiwara et al. (2017), these will be referred to as 'surface-input' reanalyses, in contrast to 'full-input' reanalyses. A third category is 'conventional-input', the sole present example being the JRA-55C product. This is noteworthy in this context as it assimilates only 'conventional', that is to say, non-satellite based, observations. It therefore provides a means of estimating of the additional value of incorporating the satellite observations. A useful comparative description of these reanalysis products including details of the underlying forecast models, the observational datasets assimilated, and the assimilation techniques used can be found in Fujiwara et al. (2017). The data used here has been re-gridded to a uniform latitude-pressure grid, and is described in Martineau et al. (2018).

Anomalies are computed from climatologies based on the years 1981 through 2001. These years are chosen since they are included in all of the reanalysis products under present consideration. Leap years are handled by omitting July 1st so that all years are treated as 365 days long. These climatologies (computed for each reanalysis) are used regardless of the period under consideration.

## 3   Sudden Stratospheric Warmings

As an initial example, Fig. 1a shows time series of zonal mean zonal wind at 60°, 10 hPa from the JRA-55 reanalysis for a set of 36 stratospheric sudden warming events, identified following Charlton and Polvani (2007). The central dates (lag 0) of the events are defined by when the wind at this grid point reverses from westerly to easterly, so all of the time series pass through zero at this point. However, the inter-event variance of the winds is large both immediately prior to and shortly after the central date. This spread is only to a weak degree the result of the timing of the event within the cold season; a similar plot of anomalies from the climatological mean shows very similar growth in the inter-event spread (not shown). As a result of this large dynamical variability, the composite mean has a large sampling variability independent of the quality of the observations or the forecast models underlying the reanalysis products.

In contrast, Fig. 1b shows the same time series from all twelve reanalysis products for a single event that occurred on 21 Feb 1989. The inter-reanalysis spread is in general much smaller than the inter-event variability emphasized in Fig. 1a. An exception to this is the surface-input reanalyses, ERA-20C and 20CR v2. JRA-55C, which does not assimilate satellite observations, is notably indistinguishable from other reanalysis products, suggesting that satellite observations are not required to closely constrain these winds.

Although there are far fewer reanalysis products that include the radiosonde period, Fig. 1c shows that the three reanalyses spanning this period which assimilate radiosonde observations (JRA-55, NCEP-NCAR, and ERA-40) exhibit a similarly close agreement, showing only a somewhat larger spread across reanalyses than in the satellite period. This again suggests that the radiosondes are providing a strong constraint on the flow, and that as a result the events that occurred during the radiosonde era are of significant potential value for constraining our knowledge of the composite mean evolution of sudden warmings.

Since sudden stratospheric warmings are typically identified by the date on which this wind reverses sign, these slight differences in reanalyzed winds can lead to the identification of central dates which differ by a day or two, and in some cases can lead to an event being identified in one reanalysis but not in others. This sensitivity is a generic feature of thresholds in the event definition, not of the particular choice of definition.

This leads to difficulties with comparing composites of events in different reanalyses: because of the large inter-event variability, the exclusion of even just one event from a given reanalysis composite mean can produce differences in the composite mean that easily overwhelm the differences in the reanalyzed flow itself. Thus small differences in the identification of events can 'alias' into relatively large apparent differences in the overall composite evolution.

Similar considerations preclude the direct comparison of composite averages of satellite-era and radiosonde-era events: they differ, but not evidently by any more than should be expected due to this dynamical sampling uncertainty. To isolate the intrinsic differences between reanalyses from this aliasing of sampling variability one must instead consider a fixed set of events across all reanalyses. This is done here by selecting the date where the event fell in the majority of the available reanalyses, following the S-RIP Chapter 6 analysis of stratosphere-troposphere coupling.

These points are illustrated in Fig. 2, which demonstrates that composites of events across reanalyses agree better when a fixed set of dates is taken than when event dates are chosen individually for each reanalysis. This is true of the full-input analyses for both the satellite era and the radiosonde era.

In contrast, the surface-input reanalyses (ERA20c and 20CR v2) generally agree better with the composites when event dates are chosen per-reanalysis, particularly around the central date of the event. This suggests that while the surface observations are sufficient to constrain the stratospheric flow to some extent, the break down of the stratospheric vortex is also significantly determined by the behaviour of the forecast model in these products.

Considering a list of fixed event dates provides a useful starting point for quantifying the additional information contained in the radiosonde era. Using the fixed set of event dates as a basis, Fig. 3a shows estimates of the overall frequency of stratospheric sudden warmings for the satellite era alone and for the full 1958-2016 era, as well as for split and displacement events. The month-by-month frequency is shown in Fig. 3b. Confidence intervals in all cases are estimated with a bootstrapping procedure: $N$ years are selected from the period from 1958 to 2016 with replacement, and the events that occurred in these $N$ years are then used to compute event frequencies, counted multiple times for those years that are selected more than once. For the satellite era $N = N_s = 32$, while for the total period $N = N_t = N_s + N_r = 53$. This whole processes is repeated 10000 times, and the bounds of the confidence intervals are taken to be the 2.5th and 97.5th percentiles.

As expected from the central limit theorem, the confidence intervals are reduced by a factor very close to $\sqrt{N_s/N_t}$. This amounts to about a 20% reduction, providing a stronger observational constraint on the climatological frequency of sudden stratospheric warmings. A similar reduction is obtained for the occurrence frequency of splits and displacements, classified following Lehtonen and Karpechko (2016), as well as for the seasonal distribution of events.

Since the bootstrapping is based on the entire record, the confidence intervals for the satellite era are not centered on the mean frequencies. The use of the longer baseline results in a slight shift of the seasonal peak, suggesting that in the long term, January events are in fact more frequent than February events, in contrast to the February peak obtained using the satellite period alone. This difference in apparent seasonality has also been discussed by Gómez-Escola et al. (2012). These changes could in principal be a result of some longer term trend or decadal variability external to the stratosphere, but they are fully consistent with the null hypothesis of sampling variability from an unchanged underlying seasonality. In this latter interpretation, the full record therefore represents a modest but useful strengthening of the observational constraints on these statistics.

## 4  Statistical Considerations

Despite these promising examples, one should expect in general that the quality of the reanalyses are not as high during the radiosonde era as during the satellite era. In this light one might regard the reduction of 20% in the confidence intervals found in Fig. 3 to be an upper bound. While errors in the reanalyses will in general arise from both observational uncertainty as well as from uncertainty arising from the underlying forecast model and assimilation process, these will be considered together here as 'reanalysis' uncertainty.

A simple way to quantify the potential improvement from including the radiosonde era is to treat the reanalysis and sampling uncertainty as uncorrelated, gaussian variance, and consider the effect on the sample mean of drawing from two periods with different variances. More explicitly, we consider some physical observable $X$ (for instance, the zonal mean zonal wind at 10 hPa and $60°$ N) to be modeled by a normally distributed random variable with mean $\mu$ and variance $\sigma^2$. Since we are interested in the statistics of the sample mean, the central limit theorem in principal allows the assumption of gaussianity to be relaxed, but the role of non-gaussian statistics will not be explicitly considered.

We further assume that the variance consists of two uncorrelated components $\sigma^2 = \sigma_d^2 + \sigma_o^2$: the first, $\sigma_d^2$, arising from the dynamical variability of the atmosphere, and the second, $\sigma_o^2$, from the reanalysis uncertainty. We further consider two sets of observations of this variable, one of $N_s$ samples with smaller reanalysis error representing the satellite era, with $\sigma_o = \sigma_s$, and one with $N_r$ samples and relatively larger reanalysis error representing the radiosonde era, with $\sigma_o = \sigma_r$. We take the dynamical variability to be constant across both samples. The variance of a sum of independent random variables is the sum of the variance of each variable; hence the variance of the sample mean during the satellite era is

$$\mathrm{Var}\left(\frac{1}{N_s}\sum_{i=1}^{N_s}X_i^s\right) = \frac{\left(\sigma_d^2 + \sigma_s^2\right)}{N_s}, \tag{1}$$

while that of the sample mean over the entire period is

$$\mathrm{Var}\left(\frac{1}{N_s+N_r}\left(\sum_{i=1}^{N_s}X_i^s + \sum_{i=1}^{N_r}X_i^r\right)\right) = \frac{N_s\left(\sigma_d^2 + \sigma_s^2\right) + N_r\left(\sigma_d^2 + \sigma_r^2\right)}{\left(N_s + N_r\right)^2}. \tag{2}$$

Here the superscript on $X$ indicates the 'era' from which the sample is drawn (and thus its variance).

A first criterion for including the both periods is that the standard deviation of the sample mean should be reduced relative to that obtained from the satellite era alone. As argued in the previous section, if the reanalysis error of the two periods are equal ($\sigma_r = \sigma_s$), the standard deviation of the mean when the whole record is considered will be reduced by a factor $\sqrt{N_s/(N_s+N_r)}$. If the reanalysis error of the two periods differ, some straightforward manipulations of the formulas above can be used to show that the factor can be written $\sqrt{N_s/(N_s+\delta N_r)}$, with

$$\delta = \frac{1 - \beta f}{1 + (1-\beta)f}, \qquad f = \frac{\alpha_r^2 - \alpha_s^2}{1 + \alpha_s^2}. \tag{3}$$

Here $\alpha_{s,r} = \sigma_{s,r}/\sigma_d$ is the ratio of the reanalysis standard deviation in each respective period to the dynamical standard deviation, and $\beta = N_s/N_t$ is the length of the satellite era as a fraction of the total length of the record. For the observational period considered here, $\beta \approx 0.6$.

The factor $\delta$ can be loosely interpreted as an efficiency factor for the sampling during the radiosonde period. Since it depends on the number of observations in both periods its value will in general change (through $\beta$) with the size of the sample; however, in the limit that the reanalysis error in both eras is small compared to the dynamical error, $\delta \approx 1 - f = 1 + \alpha_s^2 - \alpha_r^2$, in which case its value is independent of the sample size. This result, central to the argument of this work, indicates that even if the reanalysis uncertainty in the radiosonde era is much larger than the reanalysis uncertainty in the satellite era, $\delta$ will be close to 1 so long as the dynamical uncertainty dominates both.

Figure 4 shows values of $\delta$ as a function of $\alpha_r$ and $\alpha_s$ for three values of $\beta$. One can note several properties of this factor. Firstly, $\delta$ can be negative for sufficiently large values of $\alpha_r$, although this threshold depends on the value of $\beta$. For the present observational record (Fig. 4b), when $\alpha_s$ is small this occurs only when $\alpha_r$ is somewhat larger than 1, that is, when the reanalysis uncertainty is somewhat larger than the dynamical uncertainty. This threshold occurs at smaller values of $\alpha_r$ as $\beta$ decreases, so that, for marginal cases, the value of the radiosonde era in reducing overall uncertainty will decrease with time as a longer record of higher quality observations becomes available.

Secondly, $\delta$ remains close to 1 if $\alpha_r \approx \alpha_s$. Because this statistical model assumes that both periods are drawn from populations with the same underlying mean, it assigns equal value to both periods, regardless of how large the reanalysis uncertainty is relative to the dynamical uncertainty. In practice, the dynamical variability $\sigma_d$ is estimated here from the interannual variability of the field in question. The reanalysis uncertainty $\sigma_o$ is estimated from the statistics of differences between different reanalysis products: more precisely as the time mean of the standard deviation across reanalyses. If the observations are not constraining the flow in a significant way, the reanalysis product will reflect the dynamics of the underlying forecast model and the flow across the various reanalyses will become uncorrelated. In this case, assuming that the forecast models produce reasonably accurate dynamical variability, the estimate of $\sigma_o$ should approach $\sqrt{2}\sigma_d$, that is, $\alpha \approx \sqrt{2}$. To see this, consider the time series of an observable from a given reanalysis $X_i$ as the sum of the true atmospheric evolution $X_a$ and a correction $x_i$. If the standard deviation of the forecast model is correct, $X_i$ has the same standard deviation as $X_a$. When these two components become decorrelated, the correction $x_i$ will be the difference between two uncorrelated timeseries with standard deviation $\sigma_d$. Since $X_a$ is independent of the reanalysis, the standard deviation across reanalyses will therefore be $\sqrt{2}\sigma_d$.

This suggests a second criterion: if $\alpha_r$ (or $\alpha_s$) approaches $\sqrt{2}$ the observations are not providing any significant constraint on the fluctuations. In this case we should not regard the reanalysis as providing any kind of estimate of the true behaviour of the climate system and this part of the time series should not be included. To avoid influence of the forecast model, one might reasonably require $\alpha$ to be significantly less than $\sqrt{2}$.

An important assumption that has been made is that the reanalysis uncertainty is dominated by a stochastic component that is uncorrelated in time. One can easily suppose the presence of systematic errors that remain relatively fixed in time, differing only when the assimilated observations change in a substantial way. Such a systematic error will not be reduced by a larger sample size; if such an error $\epsilon$ is present during the radiosonde era, its contribution to the overall uncertainty will be $\epsilon(1 - \beta)$. However in the case that the dynamical sampling error dominates the random component of the uncertainty, this systematic error can still be neglected if $\epsilon \ll \sigma_d/\sqrt{N_t}$.

Since the dynamical standard deviation is in general a function of the flow, and the reanalysis standard deviation is a function of the observational network, the relative information content present in the radiosonde period will vary both spatially and temporally, and will depend on what quantity is under consideration. A complete survey is therefore impossible, but in the next section a brief overview of some commonly used quantities of importance to stratosphere-troposphere interaction is given.

## 5 Results

Figure 5 shows estimates of the de-seasonalized standard deviation, $\sigma_d$, and reanalysis standard deviations $\sigma_s$ and $\sigma_r$ for zonal wind in boreal winter and temperature in boreal summer. The standard deviation of the anomaly from the climatology in JRA-55 is used as an estimate of $\sigma_d$. The variability of DJF zonal winds is large in the Arctic stratospheric polar vortex, and to a lesser extent in the region of the quasibiennial oscillation (QBO) and on the flanks of the tropospheric jets. The variance of JJA temperatures also shows enhanced variance in the winter stratosphere as well as in the deep tropical stratosphere but the structures are less pronounced. In the troposphere the largest variances are at the poles.

The reanalysis uncertainty is estimated during the satellite period (Fig. 5b) as the variance across six reanalysis products (JRA-55, NCEP-NCAR R1, ERA-40, ERA-Interim, MERRA-2, and CFSR; this choice is further justified below) after first removing their respective climatological means. The variance is of the order of 0.1 m s$^{-1}$ through much of the extratropics with a slight increase with height, particularly in the winter upper stratosphere. There is considerably larger inter-reanalysis spread in the deep tropical stratosphere where the lack of strong balance constraints reduces the utility of the thermodynamic measurements available from satellites (Kawatani et al., 2016). Nonetheless the reanalysis uncertainty remains significantly less than the dynamical uncertainty throughout the QBO region, partly due to enhanced dynamical variability, and partly due the observational constraints from radiosondes. In contrast, the inter-reanalysis spread in temperatures is small (0.1 to 0.2 K) throughout most of the summer hemisphere below 10 hPa, but is larger in the upper stratosphere and the winter polar stratosphere. A weak maxima is also seen near the tropical and southern hemisphere tropopauses.

The reanalysis uncertainty during the radiosonde period (Figs. 5ef) is estimated similarly, but using the three full-input reanalyses that cover this period (JRA-55, NCEP-NCAR R1, and ERA-40). Above 10 hPa where data from NCEP-NCAR R1 is not available, the estimate is based on only two products. This results in some weak discontinuities apparent near 10 hPa. The structure of the inter-reanalysis spread is to first order similar to that during the satellite period, but is larger in magnitude. Interhemispheric differences are more apparent, with both wind and temperature spreads in general noticeably larger in the Southern Hemisphere (an exception to this is the winds in the upper stratosphere). This is generally consistent with the sparser set of observational constraints. Nonetheless in many regions it remains substantially smaller than the dynamical variability. Some features with small vertical length-scales are present in the JJA temperature variance, this is likely associated with known artificial vertical temperature oscillations present in ERA-40 (e.g., Randel et al., 2004).

The 'reanalysis' uncertainty is, as discussed above, not associated solely with the properties of the observational data available, but also of the assimilation and forecast model used by the respective reanalysis products, and could therefore depend strongly upon which products are included in the calculation. For this reason it is not immediately obvious that the inter-reanalysis spread used here is a reasonable estimate of the reanalysis uncertainty; for instance, certain reanalyses may be outliers for a given quantity and may thus inflate the overall spread.

Figure 6 thus shows pairwise inter-reanalysis differences, computed as a standard deviation over time of the difference between the anomalies from two different reanalyses. For example, if $u'_i$ is the anomalous zonal mean zonal wind of reanalysis

$i$, the difference $\sigma_{ij}$ between two reanalyses $i$ and $j$ is

$$\sigma_{ij} = \left( \frac{1}{T} \int \left( u_i'(t) - u_j'(t) \right)^2 dt \right)^{1/2}. \tag{4}$$

Entries below the diagonal are computed for the satellite period, those above the diagonal are for the radiosonde period. Entries on the diagonal show the dynamical variability computed from the corresponding reanalysis

$$\sigma_{ii} = \left( \frac{1}{T} \int u_i'(t)^2 dt \right)^{1/2}. \tag{5}$$

The ratio of the inter-reanalysis spread to the dynamical variability (an estimate of $\alpha_r$ and $\alpha_s$) are indicated by the colour of the off-diagonal cells. Red colours are chosen for ratios greater than 0.3 although this is well below the strict condition of $\alpha < \sqrt{2}$.

Differences are shown for four regions in the winters of the respective hemispheres: (a,b) in the Northern and Southern Hemisphere stratosphere (30 hPa), respectively, and (c,d) in the Northern and Southern Hemisphere troposphere (500 hPa). 30 hPa is used as a representative height for the stratosphere to reduce the effects of the model lid in NCEP-NCAR R1 and NCEP-DOE R2; otherwise the conclusions remain essentially unchanged for 10 hPa. The estimates of the dynamical variability (along the diagonal) agree closely across all reanalyses, with the exception of 20CR v2 which is significantly less variable in the stratosphere.

In the Northern Hemisphere, the agreement between full-input and conventional-input reanalyses (those other than 20CR v2 and ERA-20C) are in all cases below 30% of the dynamical variability. Looking more closely, reanalysis products that share the same or related forecast models tend to be in closer agreement than those from different centres, and there is in general better agreement between the more modern products (JRA-55, ERA-Interim, MERRA-2, CFSR) than between older products. This confirms that the forecast model and assimilation procedure is a contributing factor to the 'reanalysis' error. In the Northern Hemisphere, the agreement between the conventional-input reanalysis JRA-55C (which does not assimilate satellite observations) and other products is nearly as good as that of JRA-55, even in the stratosphere. In the Northern Hemisphere troposphere, the two surface-input reanalyses agree with other products to within 30% of the dynamical variability in the tropsphere, but this agreement degrades substantially in the stratosphere. Nonetheless, at least for ERA-20C the agreement is to within the dynamical variability, suggesting that surface observations do offer some constraint on the evolution of the stratosphere.

In the Southern Hemisphere the quality of agreement is everywhere weaker than the corresponding cases in the Northern Hemisphere. The full-input reanalyses agree to within 30% in the tropposphere, and, with a few exceptions, in the stratosphere as well. In the Southern Hemisphere, the conventional-input reanalysis, JRA-55C is more noticeably degraded relative to the agreement between other full-input reanalyses, although the differences are still substantially less than the dynamical variability. The surface-input products also show larger differences in the troposphere.

As expected, differences in the radiosonde era are in general larger than the corresponding differences in the satellite era; the one exception to this is in the Northern Hemisphere stratosphere with 20CR v2, where agreement with JRA-55, ERA-40, and NCEP-NCAR R1 are all apparently slightly improved in the absence of satellite observations. Nonetheless, agreement between these latter full-input products in the Northern Hemisphere remain very close, showing only a slight degradation within the troposphere, and an agreement between ERA-40 and JRA-55 in the Northern Hemisphere stratosphere to within 10% of the

dynamical variability. In contrast, differences in the Southern Hemisphere troposphere approach dynamical variability, and exceed it in the stratosphere.

Given the smaller sample size of products which represent the radiosonde period general conclusions cannot be as strong as those from the satellite period, nonetheless the choice of reanalyses used in Fig. 5 is justified in that no significant outliers are apparent. Lower values of the reanalysis uncertainty would likely be obtained if only more modern reanalyses were included, but this would make comparisons to the radiosonde era impossible. However, given the general improvement in agreement across modern reanalyses seen in the satellite era, it is plausible that further improvements within the radiosonde era are also possible.

Having justified to some extent the estimates of $\sigma_d$, $\sigma_r$, and $\sigma_s$, these can be used to estimate the ratios $\alpha_r$ and $\alpha_s$, and from these $\delta$ and the effective value of the radiosonde era according to the criteria discussed in the previous section. Following Fig. 5, these quantities are shown for boreal winter zonal winds and austral winter temperatures in Fig. 7.

The ratio $\alpha_s$ is seen to be in general smaller for the zonal winds than for temperatures. Consistent with Fig. 5, values are generally smallest in the Northern Hemisphere extratropics, below 0.1 for the winds and below 0.2 for temperatures. The ratio is generally below 0.4 for the winds somewhat larger values near the surface in the deep tropics as well as above 10 hPa in the tropics and at high southern latitudes. For temperatures values are below 0.4 or so in the extratropics up to about 50 hPa, but notably approach 1 near the tropopause in the tropics where dynamical variability is small, as well as in the Southern Hemisphere, and through much of the stratosphere.

The ratio $\alpha_r$ shares many of the structural features present in $\alpha_s$ but with generally larger values. Most importantly for the present discussion, the Northern Hemisphere extratropical winds show values still in general below 0.2. For zonal winds, the ratio exceeds 0.5 but remains below 1 through most of the Southern Hemisphere, indicating the observations are less effective at constraining the winds in this hemisphere, but there is still some information common across reanalyses. As with $\alpha_s$, $\alpha_r$ is larger for temperatures than for zonal winds, particularly near the tropical and Southern Hemisphere tropopause where values are well above 1. Values in the Northern Hemisphere extratropics through the lower stratosphere remain small, but the summertime mid-stratospheric temperatures (where dynamical variability is relatively weak) are not well constrained. Much of the wintertime Southern Hemisphere also shows values near 1.

Using these values of $\alpha_r$ and $\alpha_s$, Fig. 5ef show the calculated value of $\delta$. The values for the zonal wind remains quite close to 1 through the Northern Hemisphere and tropics in boreal winter. In the Southern Hemisphere below 10 hPa the values are reduced, but perhaps surprisingly remain above 0.5. This reflects to some extent the fact that the underlying reanalysis uncertainty $\sigma_s$ is larger in Southern Hemisphere than in the Northern Hemisphere, even during the satellite era. These values suggest that DJF winds are constrained well-enough by observations in the radiosonde era that they may be of some value towards reducing uncertainty. This is, however, not the case for JJA temperatures in the Southern Hemisphere (Fig. 5f, or in fact for JJA winds or DJF temperatures, though these latter cases are not shown explicitly), for which values of $\delta$ are in many cases below 0; this is notably the case for temperatures near the tropical tropopause as well.

In summary, these criteria show clear value in including the radiosonde era in dynamical analyses of Northern Hemisphere quantities from the troposphere up to the mid-stratosphere. There is a possible suggestion that useful information may be

gained for winds in the Southern Hemisphere summer winds as well. On the other hand, for much of the rest of the Southern Hemisphere quantities this is not the case. Temperatures near the tropical tropopause also show significantly worse agreement during the radiosonde period.

As they are based on the overall variance, these estimates are most sensitive to the dominant dynamical structures of inter-annual variability in the flow, which have typically relatively longer time scales and larger length scales. These bulk estimates may not therefore imply that the observational constraints on dynamical processes at shorter timescales are equally strong. To begin to assess this point, Fig. 8 compares the power spectra of deseasonalized winds from JRA-55 in the stratosphere and troposphere with the power spectra of pairwise differences between JRA-55 and other reanalyses. These provide frequency-dependent estimates of $\sigma_d$ and $\sigma_o$, respectively, and thus the ratio of these two spectra in the corresponding eras provides a frequency-dependent estimate of $\alpha_s^2$ and $\alpha_r^2$. Such spectra are shown for Northern Hemisphere winds in the stratosphere (Fig. 8a,b) and in the troposphere (Fig. 8c,d).

During satellite era differences from most reanalyses at low frequencies are two to three orders of magnitude smaller than the spectrum, consistent with the 5-10% estimate of the raw differences since these plots show the variance instead of the standard deviation. These values can be compared to the horizontal line shown at a value of 2, expected if observations are providing no constraint on the flow. Fluctuations at higher frequencies reach the same order as the dynamical variability at timescales of a few days in the stratosphere; in the troposphere differences amongst the more modern reanalyses remain below dynamical variability down to the highest frequency considered (corresponding to a period of 6 hours). Within the stratosphere differences from NCEP-NCAR R1 and NCEP-DOE R2 are significantly larger than other reanalyses at all frequencies and the differences from ERA-20C and 20CR v2 are of the order of the reference spectrum. Within the troposphere the surface-input reanalyses are still noticeably in weaker agreement with JRA-55, with difference spectra that approach the reference spectra at frequencies corresponding to periods less than half a week or so.

During the radiosonde era (Fig.8b,d) the differences are, as expected, larger than during the satellite era, although similar features can be noted with better agreement between JRA-55 and ERA-40, and significantly worse agreement with the surface-input reanalyses. This suggests that processes with timescales even as short as a few days are still significantly constrained in the Northern Hemisphere extratropics, although this constraint is not as strong (relative to dynamical variability) as is the case for processes on timescales of a month or longer.

A similar spectral analysis could be applied spatially to determine which spatial scales which are reliable. However this has not been directly considered and would be better applied to fully three dimensional data as opposed to the zonal means considered here.

Up to this point the analysis has considered both the radiosonde and satellite eras to be to some extent uniform in time in their properties; yet the observational record evolved during these periods as well. To consider briefly the evolution of the observational constraint over time, the ratio $\alpha$ can be estimated for each month individually. In this case we consider pairwise differences between JRA-55 and other reanalyses as an estimate of $\sigma_o$, and the standard deviation of JRA-55 itself as an estimate of $\sigma_d$. In all cases the time-series are first de-seasonalized.

Since the interest is primarily in the early part of the record, Fig. 9 shows this ratio for zonal winds in the Northern Hemisphere stratosphere (at 60 N, 30 hPa), and in the Southern Hemisphere troposphere (at 45 S, 500 hPa), spanning from 1958 through 1986. The month by month values fluctuate considerably, but show nonetheless a distinct annual cycle with lower values of $\alpha$ during the respective winter months when the dynamical variability is higher. A clearer trend can be observed by considering $\delta$ computed from 12-month running averages of $\alpha$ (bold lines in Fig.9). In the Northern Hemisphere stratosphere, Values for ERA-40 remain well below 0.5 through nearly all of the period in question, and NCEP-NCAR R1 is only somewhat larger. Although the methodology used here cannot yet be used to examine the period prior to 1958, these relatively low values suggest that even earlier periods could be of value. This speculation is supported by the results of Hersbach et al. (2017) who found this period to be of value in particular for constraining the evolution of the QBO.

The surface-input reanalyses show large fluctuations over time, but less of a clear trend. For ERA-20C the value of $\alpha$ remains close to 1 through much of the period, though at the beginning of the period the value is only sligly larger than for NCEP-NCAR R1. The values for 20CR v2 are systematicaly larger, not far below the limit of $\sqrt{2}$ despite the lower overall variance at these heights seen in Fig. 6.

In the Southern Hemisphere again values show a clear seasonal cycle; while there are times of the year during which the agreement is better, the 12-month runing average are above 1 for all products through the 1960s, dropping somewhat through the early 1970s and to values of less than 0.5 only after 1979. This suggests that the tropospheric flow is only weakly constrained by the observations prior to 1979. In this case the 20CR v2 shows somewhat better agreement with JRA-55 than ERA-20C through the early 1980s.

The assessment of inter-reanalysis differences presented here suggest that there is considerable value for dynamical studies in including the radiosonde era, particularly in the extratropical Northern Hemisphere. The criteria discussed suggest that for lower-frequency, large-scale processes such as those responsible for stratosphere-troposphere coupling during stratospheric sudden warmings, including the radiosonde era could reduce confidence intervals by close to 20%, despite the increase in reanalysis uncertainty during this time. To assess whether this is in fact the case, Fig. 10 presents bootstrap estimates of uncertainties (at the 95% level) on composites of several dynamical quantities fundamental to this coupling: the vertically integrated zonal wind, vertically integrated meridional momentum fluxes, and meridional heat fluxes at 100 hPa. The vertical integral is taken from 1000 hPa to 100 hPa (see, e.g., Hitchcock and Simpson, 2016). The bootstrap estimates are carried out by generating a large number of synthetic composites by selecting $N$ events with replacement from the full period (shown in solid lines with shaded confidence intervals), and from the satellite period (shown in dashed lines with outlined confidence intervals).

Importantly, any systematic error present in these quantities during the radiosonde era will contribute to the bootstrapped confidence intervals. The fact then that in each case confidence intervals are (with some regional exceptions; not shown explicitly) reduced by on the order of 20% suggests that any such systematic errors are small relative to the sampling error.

As was the case with the event frequencies shown in Fig. 3, the composite means agree nearly everywhere to within estimated confidence intervals, as should be the case. Within these uncertainties, the tropospheric jet shift is seen at somewhat lower latitudes during the full period with a less pronounced low-latitude signal; the momentum flux anomalies are somewhat more

positive, and the heat-flux anomalies during the recovery phase suggest somewhat more suppression of the upward wave flux. While the differences in composite means are modest, including this period reduces the confidence intervals on these quantities by the expected amount, providing better observational constraints on dynamical understanding and modeling efforts.

## 6  Conclusions

5    The advent of more advanced satellite-based sounding instruments in the late 1970's resulted in major improvements in the monitoring of the detailed state of the atmosphere. Nonetheless, 'conventional' upper-air observations play an important complementary role, and the network of surface and radiosonde observations in place prior to this period represent a valuable resource for observationally constraining atmospheric variability. For dynamical studies that rely on statistical composites of specific anomalous conditions, the dominant source of error in many cases arises from sampling this atmospheric variability, 10    not from observational uncertainties.

In particular, this study has considered the value of the 'radiosonde' era from 1958 to 1978 relative to the 'satellite' era from 1979 to 2010, using differences between presently available reanalysis products to characterize the constraint provided by the observations in these two periods. In principal, including the radiosonde era allows for up to a reduction of 20% in confidence intervals associated with the dynamical variability.

15    The value of the radiosonde era towards reducing the overall sampling uncertainty in composites is quantified by equation (3). This depends on the ratio of the 'reanalysis' uncertainty (including uncertainty arising both from the observations as well as that arising from the assimilation process) to the dynamical uncertainty (the variability of the dynamical phenomena themselves). A key conclusion to draw from this relationship is that even if the reanalysis uncertainty is significantly greater in the radiosonde era than in the satellite era, so long as the dynamical uncertainty dominates, the radiosonde era will be of nearly 20    equivalent value to the satellite era. However, since this criterion assesses the relative value of the two periods, it is important as well to consider directly the ratio of the reanalysis uncertainty to the dynamical uncertainty. If this is too large, this indicates a more significant influence of the underlyin forecast model.

Since these criteria depend on the physical properties of the climate system, the observations available, and of the reanalysis forecast model and assimilation system, they must be applied on a case-by-case basis. The present work cannot hope to provide 25    a comprehensive survey. However, basic zonal mean quantities including zonal winds, temperatures, and fluxes of momentum and heat, as archived for 12 reanalysis products (see Table 1) by Martineau (2017), have been considered here.

For all quantities considered, the reanalysis uncertainty in the Northern Hemisphere extratropics from the surface up to the mid-stratosphere (about 10 hPa) is found to be sufficiently small relative to the dynamical variability to make the radiosonde era of clear value in reducing composite uncertainties. For zonal mean zonal winds, the interannual variability is such that despite 30    larger reanalysis uncertainties, this is also the case for tropical winds (even in the stratosphere) and even Southern Hemisphere winds may be of some value in the austral summer. However, temperatures through much of the Southern Hemisphere are not well enough constrained to be worth including the radiosonde era. This is also notably true of temperatures in the tropical tropopause layer.

This test has also been applied to the surface-input reanalyses ERA20c and 20CR v2. The statistics of differences between these products and full-input reanalyses clearly indicate that, at least for ERA20c, their stratospheric evolution bears some meaningful resemblance to reality. However, this constraint is still much weaker to that available to full-input or even conventional-input products, with inter-reanalysis differences of similar magnitude to the dynamical variability. Furthermore, while differences between other reanalyses are reduced when considering fixed dates for stratospheric sudden warmings, for the surface-input reanalyses the comparison is improved when considering per-reanalysis dates, suggesting that, in these surface-input reanalyses, stratospheric sudden warmings are at least as much a product of the forecast model dynamics than a result of assimilated observations.

While these criteria does not consider the possibility of systematic biases in the radiosonde era, direct bootstrap estimates generally confirm this reduction in uncertainty of several dynamical quantities relevant to stratosphere-troposphere coupling following stratospheric sudden warmings in the Northern Hemisphere.

As a final note, while considerable improvements have been documented for more modern reanalyses during the satellite period (e.g., Long et al., 2017), there are at present not enough modern reanalyses that cover the radiosonde era to clearly document improvements over this earlier period. It seems likely that similar attention on the radiosonde era could produce similar improvements. Given the value of this period for dynamical studies demonstrated in this and other recent studies (Hersbach et al., 2017; Gerber and Martineau, 2018), the intent to include this period in two upcoming products (ERA-5 and JRA-3Q) is welcome.

# 7 Data availability

All analysis is based on the zonal mean dataset kindly provided by Patrick Martineau which is available online from the Centre for Environmental Data Analysis Martineau (2017).

*Acknowledgements.* The author thanks Sean Davis and Gloria Manney for helpful discussions, as well the lead authors of the SRIP chapter on Stratosphere-Troposphere Coupling, Patrick Martineau and Ed Gerber, for their support of this work. The reviewer comments of Adrian Simmons, Ed Gerber, and an anonymous referee lead to significant improvements in the text, and were also much appreciated.

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

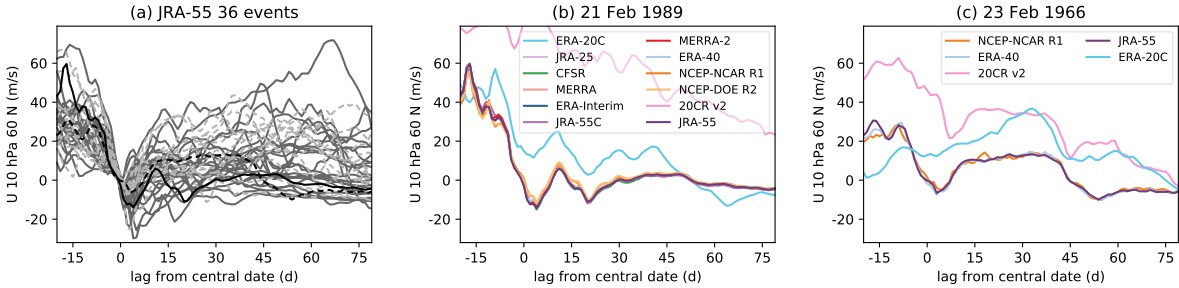

**Figure 1.** (a) Winds from JRA-55 for 36 sudden warmings. Events from the satellite period are in dark grey, those from the radiosonde period are in light grey and are dashed. (b) Winds for a single satellite-period event for all reanalyses; this event is shown by the black line in (a). (c) Winds for a single radiosonde-period event for all reanalyses covering this period; this event is shown by the dashed black line in (a).

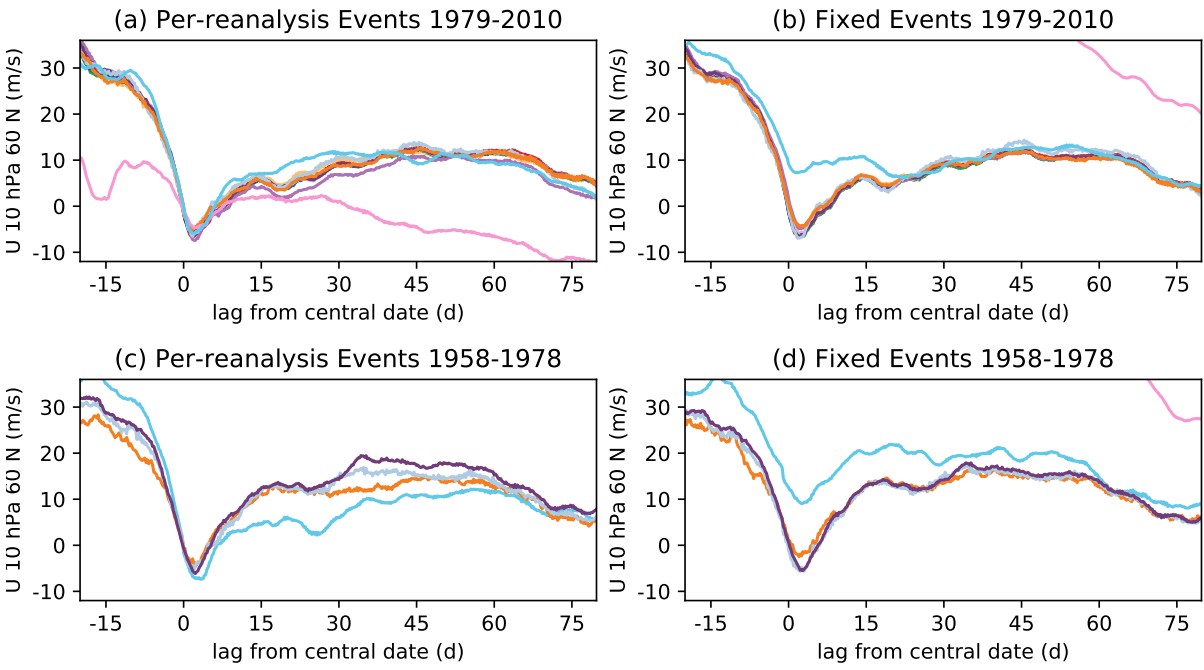

**Figure 2.** Composites of zonal mean zonal wind at 10 hPa, 60° N during stratospheric sudden warmings for events during the satellite era (a,b) and the radiosonde era (c,d). Events in (a,c) are determined by applying the wind reversal criteria of Charlton and Polvani (2007) to each reanalysis individually, while those in (b,d) are taken to be common across all reanalyses. Line colours are as in Fig. 1.

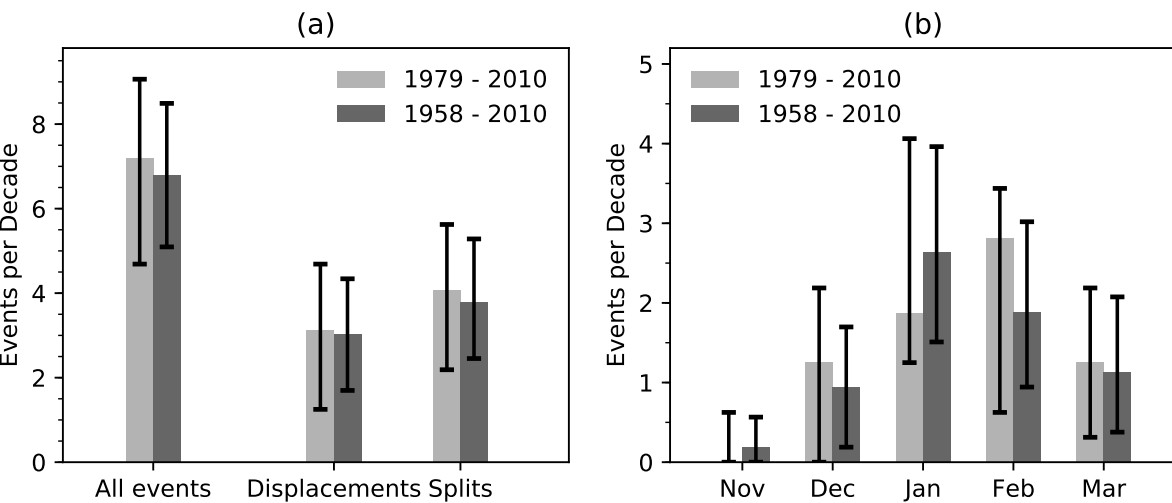

**Figure 3.** (a) Frequency of all events, and of events classified as splits or displacements for details for the satellite period versus for the radiosonde period. (b) Same as (a) but for each month of extended winter. Error bars indicate 95% confidence intervals, see text for details.

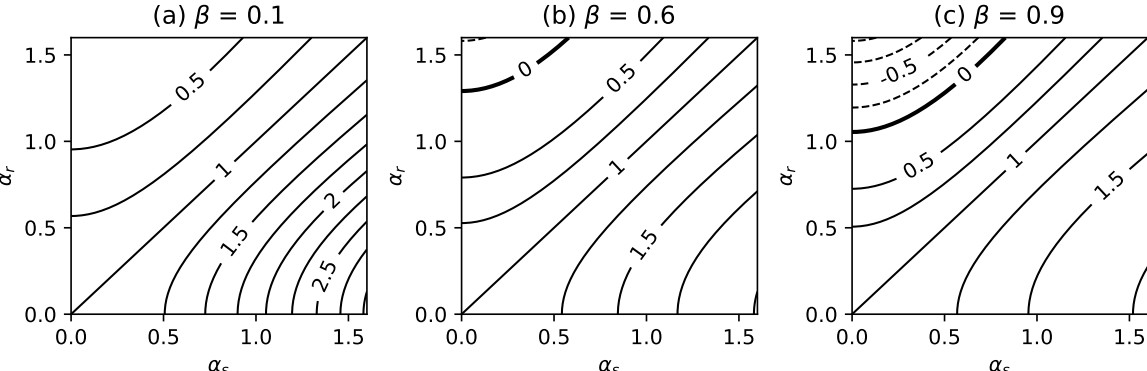

**Figure 4.** The effective value $\delta$ of radiosonde-era degrees of freedom relative to that of satellite-era degrees of freedom in reducing the overall uncertainty. Shown as a function of $\alpha_r$ and $\alpha_s$ for three values of $\beta$: (a) 0.1 (radiosonde era much longer than satellite era), (b) 0.6 (roughly appropriate for the observational records considered here) and (c) 0.9 (radiosonde era much shorter than satellite era). Contour interval is 0.25, with the 0 contour indicated in bold.

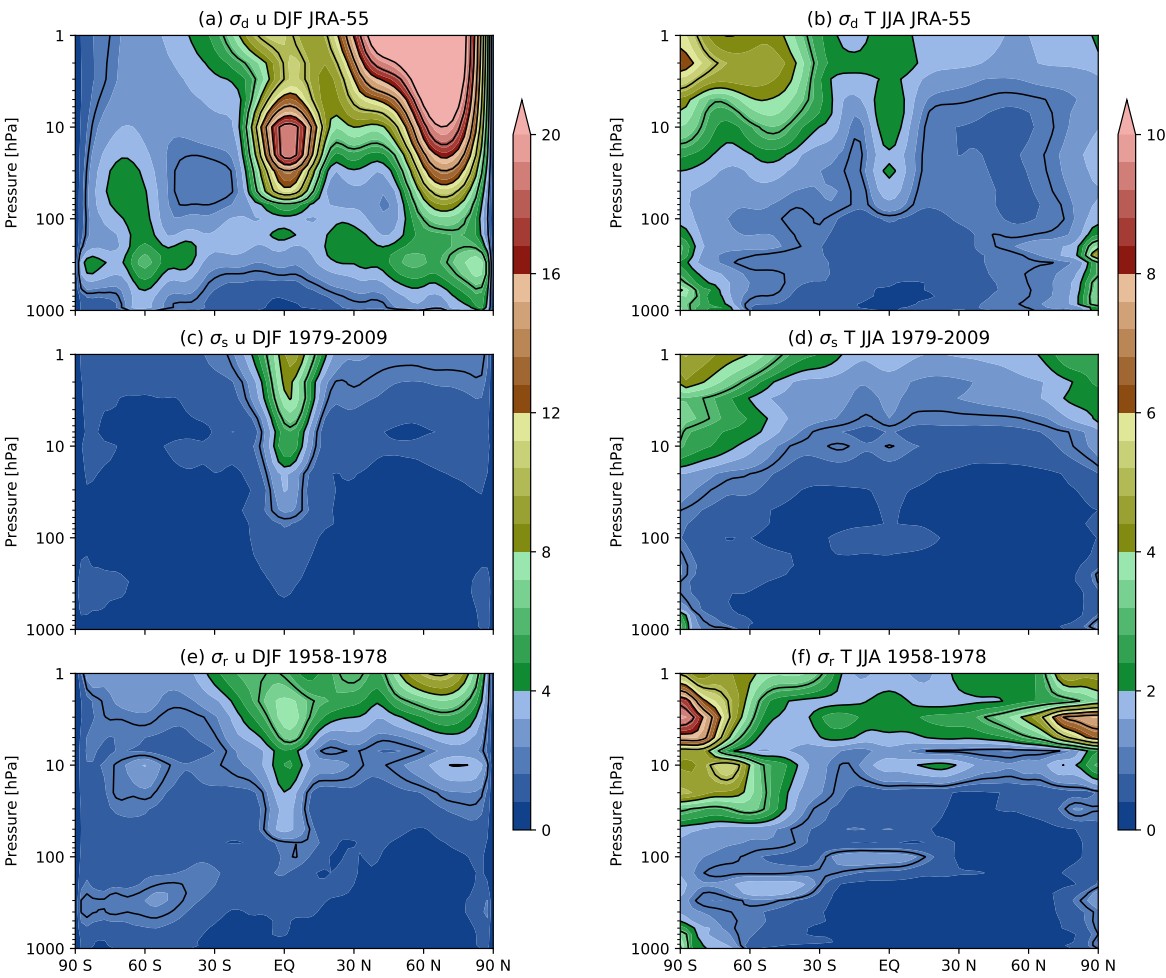

**Figure 5.** Standard deviation of de-seasonalized (a) winds in DJF and (b) temperatures in JJA from the JRA-55 reanalysis over the satellite period. (c,d) Standard deviation of the differences in same quantities (respectively) across six reanalysis products for the satellite period. (e,f) As in (c,d) but across three reanalysis products for the radiosonde period. See text for details.

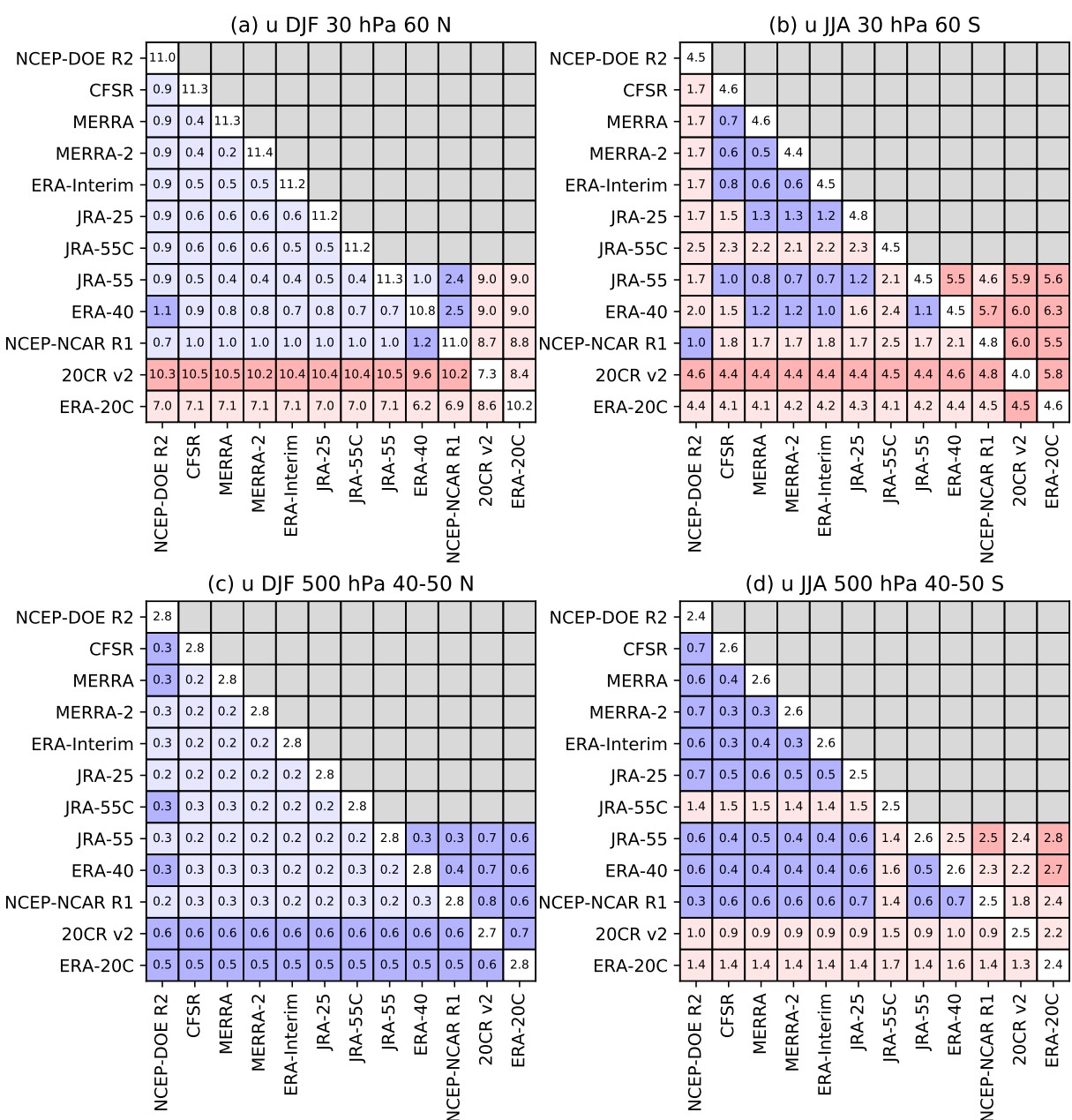

**Figure 6.** Standard deviations of pair-wise differences between winds in different reanalysis products at (a) 30 hPa, 60° N (DJF), (b) 100 hPa, 60° S (JJA), (c) 500 hPa, 40-50° N (DJF), and (d) 500 hPa, 40-50° S (JJA). All quantities are in m s$^{-1}$. The diagonal elements show the de-seasonalized standard deviation of the corresponding quantity, elements below the diagonal show differences for the satellite era, and elements above the diagonal show differences for the radiosonde era. Elements are shaded by the ratio of the difference to the mean of the dynamical standard deviations from the corresponding two diagonal elements; light blue (less than 10%), dark blue (10% to 30%), light red (30% to 100%), and dark red (greater than 100%).

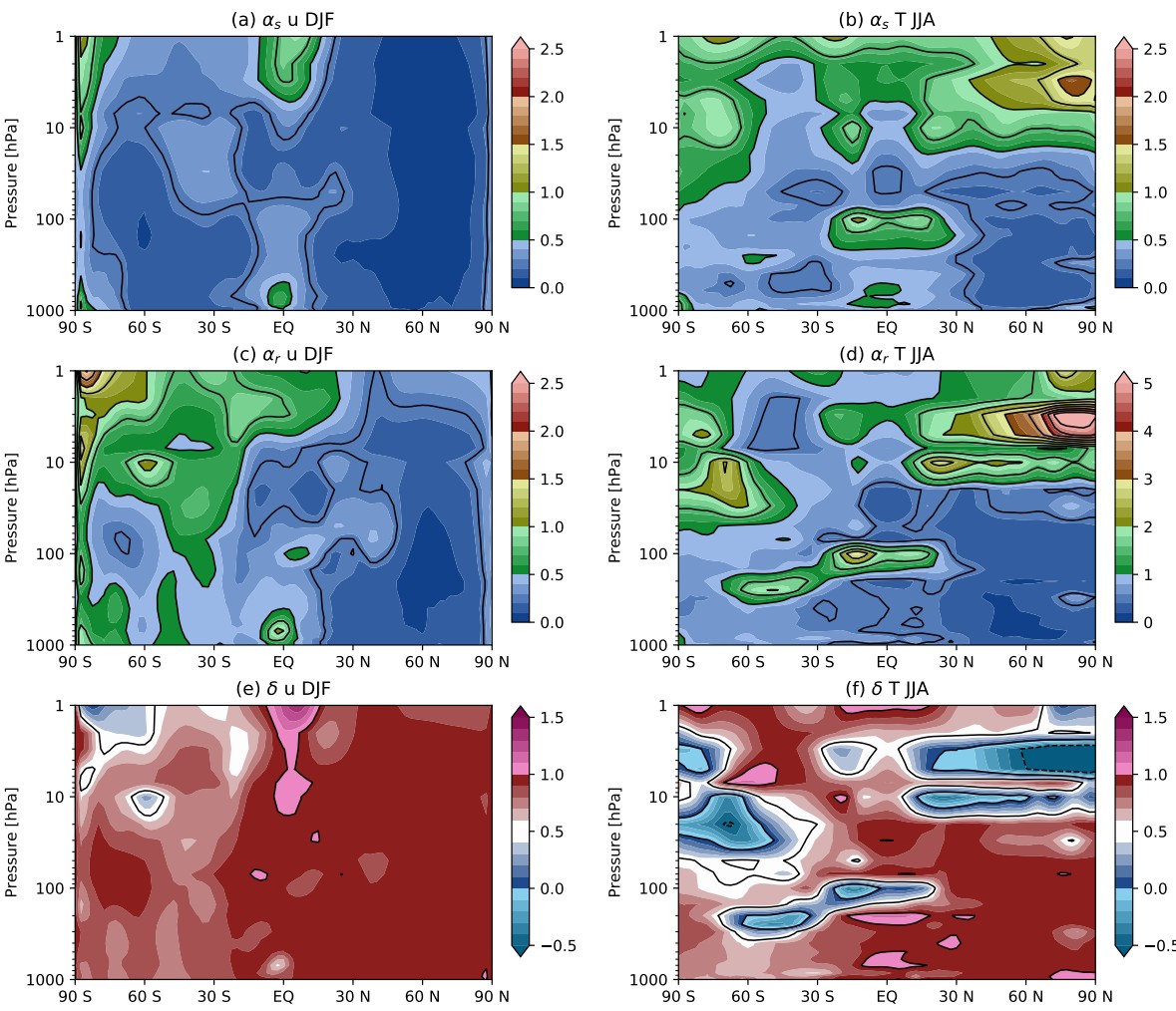

**Figure 7.** Ratios (a,b) $\alpha_s$ and (c,d) $\alpha_r$, and (e,f) the effective value $\delta$ of radiosone-era degrees of freedom as defined in Section 3 for (a,c,e) zonal winds in DJF and (b,d,f) temperatures in JJA. Note the different scale for panel (d).

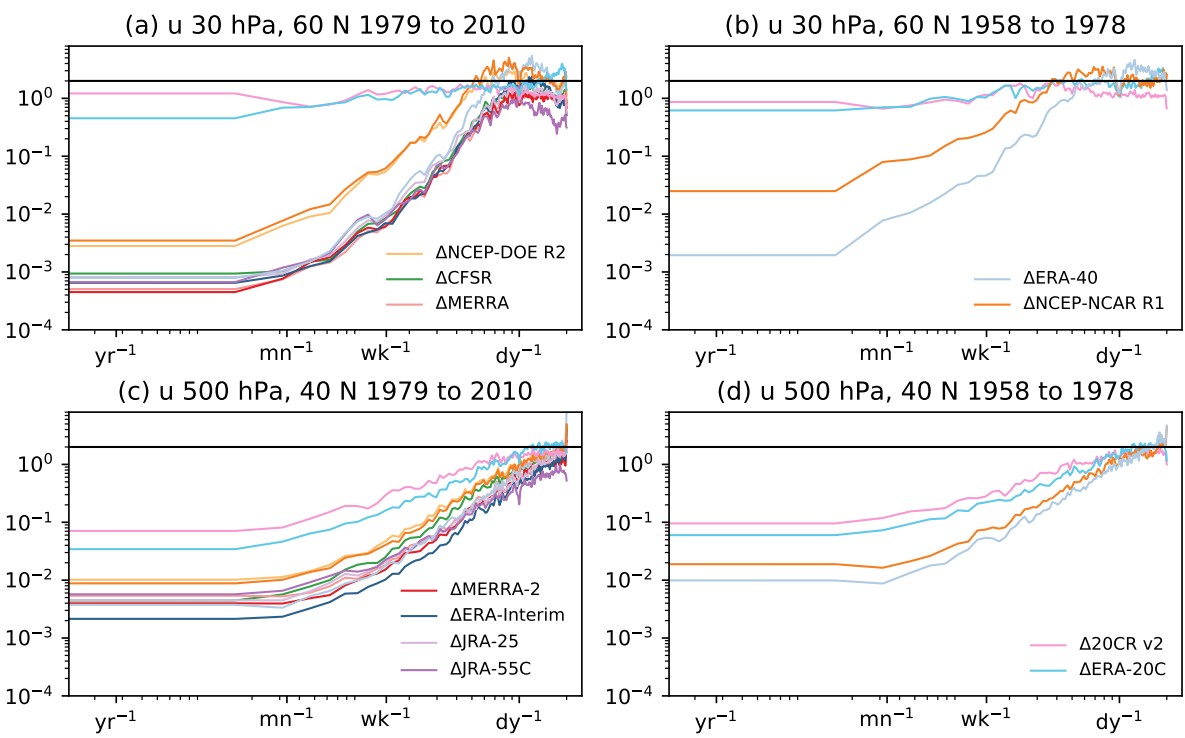

**Figure 8.** Ratio of the power spectrum of the differences in zonal winds between JRA-55 and other reanalyses (as indicated in the legend), and the power spectrum of winds in JRA-55 itself. Winds are deseasonalized and from (a,b) 30 hPa, 60 N and (c,d) 500 hPa, 40 N in the satellite era (left panels) and radiosonde era (right panels). Note that the legend is divided across the panels but applies equally to each. Frequencies corresponding to periods of one year, one month (30 days), one week, and one day are indicated on the horizontal axis. The black horizontal line is at 2, indicative of the lack of observational constraints (see text).

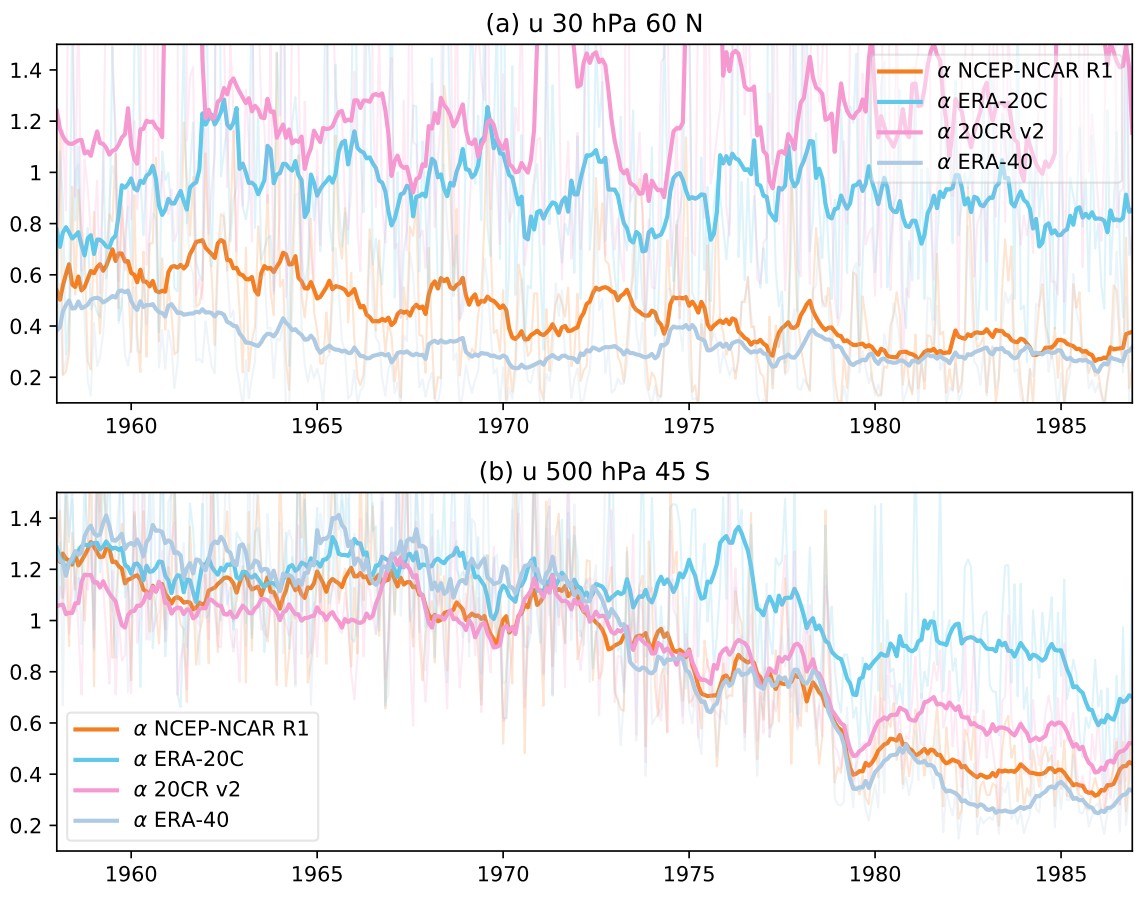

**Figure 9.** Time-dependent estimate of $\alpha$ for (a) U at 30 hPa, 60° N and (b) U at 500 hPa, 45° S. The faint lines are computed based on month-by-month variability (see text for details), while bold lines are computed based 12-month running means of $\alpha$.

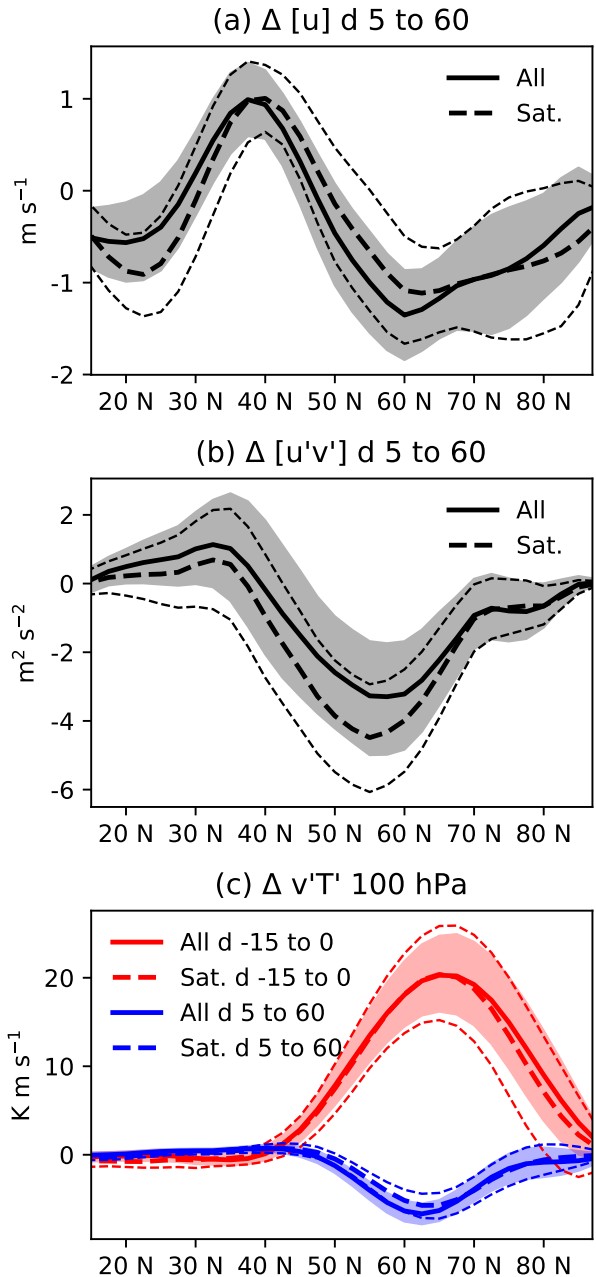

**Figure 10.** (a) Composite mean of vertically averaged zonal wind anomalies, averaged over lags 5 to 60 days following major warmings. Solid line shows the composite for all events while the dashed line shows the composite for the satellite era alone. Confidence intervals for the whole period are shaded while those for the satellite era are indicated by thin dashed lines. (b) Similar but for vertically integrated momentum fluxes. (c) Similar but for meridional heat fluxes at 100 hPa, averaged over lags -15 to 0 (in red), and over lags 5 to 60 (in blue). See text for details.

**Table 1.** Reanalysis products and dates considered in the present work. See Fujiwara et al. (2017) for a much more thorough discussion of the observations assimilated into each product. Abbreviations for certain products used within the text are indicated within parentheses.

| Product (Label) | Reference | Centre | Dates considered | Classes of data assimilated |
|---|---|---|---|---|
| JRA-25 | (Onogi et al., 2007) | JMA | 01-1979 to 12-2010 | All |
| JRA-55 | (Kobayashi et al., 2015) | JMA | 01-1958 to 12-2010 | All |
| JRA-55C | (Kobayashi et al., 2014) | JMA | 01-1979 to 12-2010 | Conventional |
| MERRA | (Rienecker et al., 2011) | NASA GMAO | 01-1979 to 12-2010 | All |
| MERRA-2 | (Gelaro et al., 2017) | NASA GMAO | 01-1981[†] to 12-2010 | All |
| ERA-40 | (Uppala et al., 2005) | ECMWF | 01-1958 to 08-2002 | All |
| ERA-Interim | (Dee et al., 2011) | ECMWF | 01-1979 to 12-2010 | All |
| ERA-20C | (Poli et al., 2013) | ECMWF | 01-1979 to 12-2010 | Surface |
| NCEP-NCAR R1 (NCEP-NCAR) | (Kalnay et al., 1996) | NOAA/NCEP and NCAR | 01-1979 to 12-2010 | All |
| NCEP-DOE R2 (NCEP-DOE) | (Kanamitsu et al., 2002) | NOAA/NCEP and DOE | 01-1979 to 12-2010 | All |
| CFSR | (Saha et al., 2010) | NOAA/NCEP | 01-1979 to 12-2010 | All |
| NOAA-CIRES 20CR v2c (20CR v2) | (Compo et al., 2011) | NOAA and CIRES | 01-1979 to 12-2010 | Surface |

[†] Although MERRA-2 includes 1980, there are spin-up issues in early 1980 which affect the Arctic vortex.