# Peer review of "On the value of reanalyses prior to 1979 for dynamical studies of stratosphere-troposphere coupling"

_Atmospheric Chemistry and Physics, 2018_

## Referee Comment (RC1) · A.J. Simmons (Referee) · 6 Nov 2018

**Review of "On the value of reanalyses prior to 1979 for dynamical studies" by Peter Hitchcock**

This paper presents an interesting and carefully presented study of the value of reanalyses covering years before 1979 for use in certain dynamical studies. It merits publication, but would benefit from minor revision to take the following comments into account.

(1) Title. As the study deals mainly with stratospheric dynamics, and stratospheric sudden warmings in particular, the author should consider changing the title so that "dynamical studies" becomes "studies of stratospheric dynamics" or "studies of the dynamics of sudden stratospheric warmings". The study does not present much evidence concerning tropospheric dynamics, nor does it reference results on tropospheric dynamics from other studies

(2) Page 1, line 3. The word "satellite" is rightly in inverted commas in the abstract. But it needs to be made clear in the body of the paper that the satellite era begins before 1979, and that it is a simplification, albeit a reasonable one, to refer to the period up to 1978 as the radiosonde era, and the period from 1979 as the satellite era. In practice (and as discussed by Uppala et al. (2005) for those observations used in ERA-40):

> (i) The MSU and SSU sounding data that characterise the start of the "satellite" era are available from November 1978.

> (ii) ERA-40 and JRA-55 assimilated radiances from the VTPR instrument available from late 1972 until early 1979.

> (iii) ERA5 is currently assimilating BUV ozone data available from 1970. Ozone analyses provide implicit information on stratospheric dynamics.

> (iv) Some cloud-tracked wind data from satellites are available and used prior to 1979.

> (v) Satellite imagery was used by the Australian Bureau of Meteorology to generate pseudo surface-pressure observations that were assimilated in ERA-40 from 1972 to 1978, although these are not being used now in ERA5 to the best of my knowledge.

In addition, there are improvements over time to the observing system, as indeed discussed in the paper under review. It is perhaps worth noting in the paper that in the "radiosonde" era, and back to the late 1940s, soundings over the North Atlantic and (to a lesser extent) the North Pacific Ocean were provided from fixed-position weather ships that were retired once satellite soundings were shown to provide a sufficient alternative. This in part compensates for lack of satellite data in the earlier years for the northern hemisphere.

(3) Page1, line 5. The word "could" should be avoided here. The paper demonstrates that the radiosonde era does extend the useful period of record back beyond 1979, so this should be made clear in the abstract. The sentence as it stands leaves the question still open.

(4) Page 1, Line 10. It is inappropriate to issue a blanket call for reanalysis centres to consider generating products prior to 1979. ECMWF did this for ERA-40, and is currently producing analyses from 1950 onwards for ERA5. ECMWF has also studied use of radiosonde and other upper-air data for the period 1939-1967 as reported in a paper by Hersbach et al. (2017, doi: 10.1002/qj.3040) that rather surprisingly is not referenced in the paper under review. JMA ran JRA-55 from 1958 onwards, and will soon start production of JRA-3Q, for which the plan is to start in the late 1940s. So these two major producers appear already to appreciate the value of products prior to 1979 – though further evidence as provided by the paper under review is always welcome.

(5) Page 2, line 8. It would be appropriate here to record that ECMWF is currently producing ERA5 reanalyses from 1950 onwards and that analyses from the late 1940s onwards are expected from JRA-3Q.

(6) Page 2, line 18. A reference to Hersbach et al. (2017) could be introduced here.

(7) Page 2, line 22. It could be referenced here that Simmons et al. (2005, J.Atmos. Sci, March) demonstrated that the ERA-40 reanalysis was of sufficient quality in January 1958 to produce a good five-day forecast of the split-vortex sudden warming that occurred during that month. Caveats were issued in this paper about the quality of the stratospheric analyses over the southern hemisphere prior to 1979, but these analyses nevertheless gave no indication of a major split-vortex event between 1957 and 1978 of the type observed in September 2002, a result consistent with analysis of the sparse radiosonde data available for the period.

(8) Page 5, lines 12 to 14. The text here needs revising. It refers to "uncertainties" in observations but "errors" in forecast models and the assimilation process. In reality there are errors in observations, and uncertainties in modelling and assimilation due to the stochastic nature of some of the processes being dealt with. So one should not use one word for observations and another for models/assimilation.

(9) Page 5, equation (2). The upper limit of the second sum on the left-hand side of the equation should be $N_r$ not $N_s$.

(10)  Page 6, lines 21 and 22. Same comment as (8) regarding the use of the words "observational uncertainty" and "errors in the forecast model and the assimilation process".

(11) Page 7, line 6. The text on Fig5 refers to $\sigma_{sat}$ and $\sigma_{rad}$, whereas Fig 7 and the text refer to $\sigma_s$ and $\sigma_r$. This should be rectified.

(12) Page 7, line 16. The lack of a strong balance constraint is a reasonable explanation for the reanalysis uncertainty in the tropical upper stratosphere. But reanalysis uncertainty is much lower than dynamical variability at 10hPa and below. This is presumably because radiosonde data alone are quite effective in constraining the QBO in the lower and middle stratosphere in reanalyses. A comment could be added to this effect.

(13) Page 7, lines 17 and 18. Manney et al. (2005) is the reference the author chooses to use here. But the deficiency of ERA-40 under discussion was first identified in preparing a SPARC Report on stratospheric climatology, and this was published in a subsequent peer-reviewed paper by Randel et al. (2004), i.e. earlier than the Manney et al. paper. So Randel's paper would probably be a fairer reference. The problem was also acknowledged in Uppala et al.'s (2005) write-up of ERA-40.

(14) Page 10, line 26. Delete the word "at".

(15) Page 10, line 34. This is another place where a reference to Hersbach et al. (2017) could be added, as that paper discusses, *inter alia*, the utility of 1950s radiosonde data for analysing the QBO.

(16) Page 11, line 23. "following 1979" should at least be changed to "following 1978" and more precisely could be written "from late 1978 onwards". The subsequent reference to radiosondes being remarkably effective in constraining the boreal stratosphere from 1958 to 1978 perhaps can remain as is, in view of the results of JRA-55C, even though VTPR data provide an additional constraint from late 1972.  It perhaps should be recognised however that radiosonde observations continue to provide a constraint on the stratosphere from 1979 onwards. Satellite radiances (particularly from the TOVS instruments flown from 1978 until phased out between 1998 and 2006)

have significant biases, and radiosondes play an important role in the bias-correction schemes for radiance data used by reanalysis centres, at least prior to the availability of substantial amounts of GPS radio occultation data from 2006 onwards. The better-quality reanalyses produced for the period from 1979 onwards is due to the combined use of radiosonde and satellite data, notwithstanding the labelling of the period as the "satellite era".

(17) Page 12, line 15. "be" should be "been".

(18) Page 12, line 21. Some rewording is required here, as it is a bit misleading to categorize the sudden warmings in full-data reanalyses as a "result of assimilated observations". They are a result of assimilating observations making use of a forecast model, and as such are a result of both forecast-model dynamics and assimilated observations. The reanalyses that assimilate only surface observations demonstrate that assimilating upper-air observations is important, but does not show that the forecast model is unimportant.

(19) Page 12, lines 22 to 25. Newer reanalyses apply bias corrections to radiosonde data (generally following the work of Haimberger), and assimilating the bias-corrected radiosonde data tends to control the biases of the reanalyses, at least in places and at levels radiosonde data are reasonably plentiful. No bias-correction of pre-1979 radiosonde data was applied in ERA-40, but the radiosonde data would nevertheless have limited systematic error in ERA-40 to some extent. A change is not called for at this point in the paper, but consideration could be given to writing something earlier in the paper on this point.

(20) Page 12, lines 29 and 30. Comment (4) above, concerning the final sentence of the abstract, applies equally to this final sentence of section 6.

---

## Referee Comment (RC2) · Anonymous Referee #3 · 9 Nov 2018

**Review of the manuscript "On the value of reanalyses prior to 1979 for dynamical studies" by P. Hitchcock**

**General Comments**

The author presents a detailed comparative analysis of the quality of reanalyses data prior to 1979 and their potential inclusion in dynamical studies. In particular, he focuses on the analysis of relevant fields for the stratosphere-troposphere coupling. The results indicate that reanalysis data in the pre-satellite era is of sufficiently high quality to be considered together with the data of subsequent decades in these dynamical studies.

The manuscript is well written and the topic is certainly interesting for the scientific community, particularly that focused on stratosphere-troposphere coupling. The methodology applied for evaluating the quality of reanalysis data is also very thorough. However, in some cases I find the text a little bit dense particularly when describing Figures 6 and 7 and Section 4 and it would be advantageous for the manuscript to try to simplify that description. Thus, I recommend the publication of the manuscript after the mentioned minor correction and some other slight changes indicated below.

**Specific Comments**

Page 5 Lines 5-8: The author indicates that the shift of the seasonal peak of SSWs in the satellite era with respect to the whole period (1958-2010) is only just due to the consideration of a longer database. I think the author could discuss a little bit more about this. Otherwise, the reader might get the impression that this is only a possible bias due to the lack of assimilated satellite data in the pre-satellite period. In contrast, it could be also related to multidecadal climate variability. Indeed, there are some studies that have also shown a change in the seasonality of SSWs in model simulations (e.g.: Ayarzagüena et al. 2013). Finally, I would recommend citing here Gómez-Escolar et al. (2012) that already showed the change in the seasonal distribution of SSWs between the pre- and post-satellite periods.

Page 7 lines 23-25: Maybe I am getting something wrong but the largest spreads, at least for the zonal wind, are found in the Northern Hemisphere.

Page 8 lines 11-12: Please indicate why you are selecting different levels for the stratospheric field in the Northern and Southern Hemisphere.

Page 9 lines 20-25: I think the author should be careful with the description of the results in this paragraph. For instance, some fields that are indicated as not shown ($\delta$ for T in JJA, ($\delta$ for DJF u) are in fact shown and some others described as shown are not ($\delta$ for T in DJF, ($\delta$ for u in JJA). It would also help if a reference to the plots is included in each case too.

Page 10 lines 32-35: I might agree that data of 1950s may be of interest, but the results for NCEP/NCAR reanalysis for that decade are not shown in Figure 9.

**Technical comments**

Page 4 Line 21: then → than
Page 4 Line 31: I think it would be better to write "from 1958 to 2016".
Page 5, line 15: means → mean.

Page 5, equation 2: in the second sum the upper limit should be $N_r$ instead of $N_s$.
Page 5 equation 2: Please define $N_t$
Page 7 line 15: I think it is the winter ***upper*** stratosphere.
Page 7 line 25: in many regions in → in many regions it
Page 8 line 25: Southern Hemisphere
Page 10 line 26: Please delete at.
Page 10 line 30: Please include ) after 9.
Page 11 line 6: reduced → reduce.

---

## Referee Comment (RC3) · E. P. Gerber (Referee) · 9 Nov 2018

This is an interesting and well thoughout study about the potential value of earlier, pre-satellite era reanalysis records. It is important to quantify the potential value of this earlier period, as it is a major undertaking for a reanalysis center to provide pre-satellite reanalyses. With the exception of JRA-55 (and the ERA5 analysis, currently in production), most of the state-of-the-art full input reanalyses do not begin until 1979 (ERA-I, MERRA, CFSR) or 1980 (MERRA2).

I recommend publication of the manuscript pending consideration of the comments below. They are mostly minor, in that I leave them to the author's discretion, but I hope that responding to them would improve the impact of the paper. (An exception is that

the author does need to better define a few things, to ensure the results are reproduce-able. But this will be easy to do.) My more philosophical question about the proposed metric for assessing the value of earlier reanalyses (see below) is perhaps trickier to fully answer, and might be something for future work. I think that the contributions of this paper are already worthy of publication. Given that it could be a subject for future research, I'll sign this review, as I would welcome discussion with author. Edwin Gerber

General comments

1) A few key elements of the procedure were not sufficiently documented. In particular, how were the SSW dates set, and how were the events classified in the spilits or displacements. I suspect this was done within the S-RIP Chapter 6 framework, assembled by Amy Butler. If so, I am not sure how to properly cite this information at this time, though they will be published. In any case, to reproduce these results, the reader does need to know the dates, and some insight on how they were obtained.

2) It would help the reader to adopt a consistent use of the nomenclature "full-input", "conventional-input", and "surface-input" throughout the paper. I appreciate that terms evolved in parallel to this research, but as a result of this time mismatch, they appear inconsistently through the text.

3) I very much appreciate the central result of the manuscript: equation (3) and surrounding discussion, which seeks to quantify the value of earlier records. I was admittedly surprised, however, that the metric indicates that there is considerable value to much of the data in the austral hemisphere, where we know that the large scale circulation is not consistently captured by the reanalyses. (In Gerber and Martineau, 2018, for example, we found that the southern annular mode indices in JRA-55, ERA-40, and NCEP-R1 share only a small fraction of the variance during the pre-satellite period, indicating that there is very little consensus on the large scale state of the austral hemisphere on synoptic time scales.)

I think the key is the assumption that reanalyses properly capture the dynamical uncertainty, $\sigma_d$, in both the satellite and pre-satellite periods. I think this effectively implies that we trust their climatological values and variance, even if the become untethered to observations.

To make my concern clear, consider the extreme case where the reanalyses are perfect in the satellite era ($\alpha_s \to 0$) and know absolutely nothing about the state of the atmosphere in the radiosonde era ($\alpha_r \to \sqrt{2}$). In this case, f-> 2 and $\delta$ -> [1-2*$\beta$]/[1+(1-$\beta$)*2]

When $\beta$ becomes small (<0.5), you would still conclude that there is value in the reanalysis, even though it knows nothing about the state of the atmosphere. (The "real" $\beta$ is about 0.6, so in this limiting case delta would be negative, and you would concluded there is no value in earlier records). But given that $\alpha_s$ is not zero, and there is some limited skill in the radiosonde era, it's not hard to see why delta is positive. And by this logic, there would be considerable value in using the entire record from ERA-20C, where beta drops below 0.5!

My intuition if we want an *observationaly constrained* estimate of the uncertainty, then we should only include the information from the earlier period when $\alpha_r < 1$. That is, when uncertainty in the renanalyses reaches the level of dynamical uncertainty, then we can argue the reanalyses are sufficiently untethered from the real atmosphere to provide any additional information than you could obtain from simply running a forecast model untethered to observations.

I haven't thought this through enough to provide a good way to quantify the value of events when $\alpha_r < 1$. It helps me to think of this interms of events (as with the SSW composites shown by the author.) Suppose you have N events from the satellite record. Looking at past events, the idea would be to quantify the additional information content of each radiosonde period event on an event-by-event basis. When the spread between reanalyses for an earlier event is equivalent to the spread between events in the satellite period ( $\alpha_r = \alpha_s$), the event is clearly of complete value

($\delta=1$); it should be added fully. Now your composite is based on N+1 events, and the uncertainty drops accordingly.

If the spread between the reanalyses for the event, however, becomes equivalent to the climatological/dynamical spread ($\alpha_r = 1$) then I feel that there's no additional information to be gained than if you simply ran a free running model: this event should be given zero value. I am just not sure how to develop a meaningful way to interpolate inbetween these cases.

4) I appreciate that the comment above is weak on specific suggestions. To be more concrete, I would have appreciated more discussion of the different limits around equation (3). The limit where $\alpha_s$ is small and $\alpha_r$ approaches sqrt(2) was interesting to me, as it drove home this issue of whether we ought to trust a good model that is untethered to reality.

Another problematic limit is $\alpha_r = \alpha_s$. Here, you always use more data, even if it's all untethered to reality. (Based on my arguements above, the value of the reanalyses should be zero when $\alpha_r$ or $\alpha_s$ approaches 1.)

And not to be overly critical, Figure 4 was not easy to interpret. Consider using color or more simply marking the contours. (I know that I should have realized that diagonal is 1 by definition, but it took me time at first reading.) I also think that it's inappoatiate to show such a range. Once $\alpha_s$ or $\alpha_r$ reach sqrt(2), nothing is tethered to observations, and I don't see how $\delta$ is meaningful for values beyond this point.

5) There is a paper that can be cited for the Martineau data set:

Martineau, P., Wright, J. S., Zhu, N., and Fujiwara, M.: Zonal-mean data set of global atmospheric reanalyses on pressure levels, Earth Syst. Sci. Data, 10, 1925–1941, doi:10.5194/essd-10-1925-2018, 2018b.

Also, it's my understanding that MERRA2 has a DOI that should be cited, as it's very important for them to justify resources. I find the situation problematic, in that you got

the data from a different source (which did cite this doi), but perhaps you could add the doi to the data section.

Global Modeling and Assimilation Office: MERRA-2 inst3_3d_asm_Np: 3d, 3-Hourly, Instantaneous, Pressure-Level, Assimilation, As- similated Meteorological Fields V5.12.4, https://doi.org/10.5067/QBZ6MG944HW0, http://disc.sci.gsfc.nasa.gov/mdisc/, Goddard Earth Sciences Data and Information Services Center (GES DISC), Accessed: 2017-07-5, 2015.

Small comments by page:line

2:3-8 This would be a good time to differentiate and define full-, conventional- and surface-input reanalyses.

2:21 \citep[e.g.,][] (Also, I think that perhaps one should include the comma on e.g., since if you spelled out the phrase, it would be: for example, Matsuno 1971. But perhaps this is a case where American English is different from British.)

2:34 I might break this off as a full sentence, instead of using the semicolon.

3:11 I appreciate why the author states that they are constrained *primarily* by surface observations (as the reanalyses are given changes in radiative gases, etc.), but this sentence seemed a bit to vague.

3:25 consider rephrasing this sentence: I understood it completely, but had to re-read it a few times

4:16 This would be a place to explain how the dates were set, and how splits and displacements were classified, or at least point the reader to the necessary information.

4:24-26 This is a fascinating/perplexing result. I think it makes sense, though: ERA-20C does a good job of getting SSWs, but since it only gets the dates right for half of them, you are better off treating it as a free running model (i.e. not fixing dates to reality) than trying to make it conform with what actually happened in our atmosphere. ERA-

20C provides the challenge to your metric in equation 3: I do think you would argue that it's worth while using the entire record, even if just assume it knows nothing about the actual state of the atmosphere. That's what motivated my thoughs on comment 3 above.

4:29 splits and displacements need to be defined

5:1 second half of the line is awkwardly phrased

5:17 I'm concerned that the zonal mean wind at 60 N and 10 hPa is decidely not Gaussian, and rather skewed towards negative values.

5:18. There is a sentence between when you introduce \sigma_d and _o and define them. Consider moving the first sentence of the next paragraph up, to define the variables, before discussing the central limit theorem.

6:10 In the limit where the reanalysis error is small relative to the dynamical uncertainty, isn't f small, and delta about equal to 1?

8:20-24 This sentence is long. Consider breaking at the ;, and then being more clear what agreement you mean to refer to.

Fig. 8: I assume the Fourier analysis was done on the deaseasonalized winds, as there's no discernable annual cycle peak here!

10:21 and Figure 7 e,f I am confused how you can estimate \delta without knowing \beta. It only seems to decouple from \beta when f is small. And in this limit, \delta would be close to 1 (and it's sort of a trivial result: you trust everything.)

In the figure, the value of delta varies considerably (changing sign!) so you must have some finite value of \beta. What is 0.6?

11:1-2 This sentence could be split up, giving you two sentences, enough to justify a paragraph!

11:13-15 It was hard for me to see this important result. At some latitudes (c. 65 in panel a of Fig 10, or near 75 in panel b), the uncertainty bounds on the full record were larger than than for the satelite record. So clearly there wasn't always a 20% reduction.

I'm was also rather struck by the fact that the dashed curved in panel b of Figure 10 approaches the edge of the confidence interval on the "all" composite. Does this mean that they were almost statistically different, or would this only apply when the confidence intervals themselves separate.

From a practical standpoint, if I wanted to ask whether my model was signficantly different from our best estimate of observations, which error bound should I use?

11:30 Might be good to emphasize "has been quantified in equation (3)."

12:2 Related to some comments above, when the dynamical uncertainty dominates, doesn't this imply that you trust everything?

12:15 An opportunity to use surface-input nomenclature.

15:18 I am not sure how to see this in Figure 9. Doesn't the fact that $\delta$ is consistently greater than 0 for ERA-20C imply that there's always value to be found from this reanalysis?

15:26-30 Could be opportunity to highlight that your message has been heard, and ERA5 hopes to go back to 1950.

Fig 4: See my general comment (4) above. I think this figure could be improved.

Fig 5: Your notation differs a bit here, $\sigma_d$ vs. $\sigma_{dyn}$. It's clear enough for the reader, but consistency is best.

Fig 7: caption has the wrong symbol. I would have appreciated more detail here in how the bottom panels were computed.

Fig. 8: I find that the log scale makes comparison very difficult. Would it be possible

to show the ratio of the differences in the power spectra? This is a number that would presumable vary from 0 to about 2 for all timescales. (It would be 2 if the limit that the models become untethered from observations. I guess it could become larger if there are systematic biases.)

If nothing else, the reference time series of JRA-55 gets buried by the other lines: consider bringing it up to the top. (If you produced this plot with matlab, but want to keep it first in the legend, a solution is to just print it again.)

Figure 10: Are these 95% confidence intervals?

---

## Author Comment (AC1) · 18 Jan 2019

*I thank all three reviewers for their considered, constructive comments. They have significantly improved the manuscript. I have included my detailed responses in italics inline below.*

**Response to Reviewer 1**

This paper presents an interesting and carefully presented study of the value of reanalyses covering years before 1979 for use in certain dynamical studies. It merits publication, but would benefit from minor revision to take the following comments into account.

(1) Title. As the study deals mainly with stratospheric dynamics, and stratospheric sudden warmings in particular, the author should consider changing the title so that dynamical studies becomes studies of stratospheric dynamics or studies of the dynamics of sudden stratospheric warmings. The study does not present much evidence concerning tropospheric dynamics, nor does it reference results on tropospheric dynamics from other studies

*The focus is on stratosphere-troposphere coupling - Fig.5 through 10 all include some component of tropospheric dynamics, and one of the points made is that many of the open questions in this area revolve around tropospheric feedbacks (for which the momentum fluxes considered in the final figure play a central role). The suggested titles would thus mischaracterize the text. Moreover, the criteria developed for evaluating this period are just as applicable to other composite-based dynamical studies (for example); this was the justification for the title. Still, given the emphasis on stratosphere-troposphere coupling, I have added this to the title.*

(2) Page 1, line 3. The word satellite is rightly in inverted commas in the abstract. But it needs to be made clear in the body of the paper that the satellite era begins before 1979, and that it is a simplification, albeit a reasonable one, to refer to the period up to 1978 as the radiosonde era, and the period from 1979 as the satellite era. In practice (and as discussed by Uppala et al. (2005) for those observations used in ERA40):

(i) The MSU and SSU sounding data that characterise the start of the satellite era are available from November 1978.

(ii) ERA40 and JRA55 assimilated radiances from the VTPR instrument available from late 1972 until early 1979.

(iii) ERA5 is currently assimilating BUV ozone data available from 1970. Ozone analyses provide implicit information on stratospheric dynamics.

(iv) Some cloudtracked wind data from satellites are available and used prior to 1979.

(v) Satellite imagery was used by the Australian Bureau of Meteorology to generate pseudo surfacepressure observations that were assimilated in ERA40 from 1972 to 1978, although these are not being used now in ERA5 to the best of my knowledge.

In addition, there are improvements over time to the observing system, as indeed discussed in the paper under review. It is perhaps worth noting in the paper that in the radiosonde era, and back to the late 1940s, soundings over the North Atlantic and (to a lesser extent) the North Pacific Ocean were provided from fixedposition weather ships that were retired once satellite soundings were shown to provide a sufficient alternative. This in part compensates for lack of satellite data in the earlier years for the northern hemisphere.

*Thank you for these details. I have added further clarification of the use of 'satellite' and 'radiosonde' eras as convenient simplifications, emphasizing in the introduction both the availability of satellite data prior to 1979 as well as the availability of radiosonde data prior to 1958, and their continued importance after 1979. I have however avoided getting too much into the history of the observational network as I don't feel I have expertise or historical knowledge to do this justice.*

(3) Page1, line 5. The word could should be avoided here. The paper demonstrates that the radiosonde era does extend the useful period of record back beyond 1979, so this should be made clear in the abstract. The sentence as it stands leaves the question still open.

*This is a good point. The text has been changed as suggested.*

(4) Page 1, Line 10. It is inappropriate to issue a blanket call for reanalysis centres to consider generating products prior to 1979. ECMWF did this for ERA40, and is currently producing analyses from 1950 onwards for ERA5. ECMWF has also studied use of radiosonde and other upperair data for the period 19391967 as reported in a paper by Hersbach et al. (2017, doi: 10.1002/qj.3040) that rather surprisingly is not referenced in the paper under review. JMA ran JRA55 from 1958 onwards, and will soon start production of JRA3Q, for which the plan is to start in the late 1940s. So these two major producers appear already to appreciate the value of products prior to 1979 though further evidence as provided by the paper under review is always welcome.

*I have added citations and discussion of Hersbach et al. in several appropriate places (see also responses below), thank you for bringing this work to my attention. I have left in the recommendation that future reanalyses include this period as I see no reason not to do so. Of course reanalyses centers are responding to the needs of a huge range of users who benefit greatly from this service, and there are accordingly a wide diversity of priorities. My intent with this paper was to put on record a quantitative argument for why this period is of value, partly in the hope that it can be useful to reanalysis centers in justifying their use of resources.*

(5) Page 2, line 8. It would be appropriate here to record that ECMWF is currently producing ERA5 reanalyses from 1950 onwards and that analyses from the late 1940s onwards are expected from JRA3Q.

*This has been done.*

(6) Page 2, line 18. A reference to Hersbach et al. (2017) could be introduced here.

*I have introduced it a bit later in the introduction.*

(7) Page 2, line 22. It could be referenced here that Simmons et al. (2005, J.Atmos. Sci, March) demonstrated that the ERA40 reanalysis was of sufficient quality in January 1958 to produce a good fiveday forecast of the splitvortex sudden warming that occurred during that month. Caveats were issued in this paper about the quality of the stratospheric analyses over the southern hemisphere prior to 1979, but these analyses nevertheless gave no indication of a major splitvortex event between 1957 and 1978 of the type observed in September 2002, a result consistent with analysis of the sparse radiosonde data available for the period.

*A citation to Simmons et al. 2005 has been added here.*

(8) Page 5, lines 12 to 14. The text here needs revising. It refers to uncertainties in observations but errors in forecast models and the assimilation process. In reality there are errors in observations, and uncertainties in modelling and assimilation due to the stochastic nature of some of the processes being dealt with. So one should not use one word for observations and another for models/assimilation.

*I have reworded these and other sentences to avoid associating 'error' or 'uncertainty' specifically with the observations or the modelling/assimilation process.*

(9) Page 5, equation (2). The upper limit of the second sum on the lefthand side of the equation should be N r not N s .

*Changed - thank you for noticing this.*

(10) Page 6, lines 21 and 22. Same comment as (8) regarding the use of the words observational uncertainty and errors in the forecast model and the assimilation process.

*Reworded.*

(11) Page 7, line 6. The text on Fig5 refers to  sat and  rad , whereas Fig 7 and the text refer to  s and  r . This should be rectified.

*Fixed.*

(12) Page 7, line 16. The lack of a strong balance constraint is a reasonable explanation for the reanalysis uncertainty in the tropical upper stratosphere. But reanalysis uncertainty is much lower than dynamical variability at 10hPa and below. This is presumably because radiosonde data alone are quite effective in constraining the QBO in the lower and middle stratosphere in reanalyses. A comment could be added to this effect.

*This is a good point and has been commented on in the updated text.*

(13) Page 7, lines 17 and 18. Manney et al. (2005) is the reference the author chooses to use here. But the deficiency of ERA40 under discussion was first identified in preparing a SPARC Report on stratospheric climatology, and this was published in a subsequent peerreviewed paper by Randel et al. (2004), i.e. earlier than the Manney et al. paper. So Randels paper would probably be a fairer reference. The problem was also acknowledged in Uppala et al.s (2005) writeup of ERA40.

*The reference has been changed to Randel et al. 2004.*

(14) Page 10, line 26. Delete the word at.

*Done.*

(15) Page 10, line 34. This is another place where a reference to Hersbach et al. (2017) could be added, as that paper discusses, inter alia, the utility of 1950s radiosonde data for analysing the QBO.

*Agreed - this has been done.*

(16) Page 11, line 23. following 1979 should at least be changed to following 1978 and more precisely could be written from late 1978 onwards. The subsequent reference to radiosondes being remarkably effective in constraining the boreal stratosphere from 1958 to 1978 perhaps can remain as is, in view of the results of JRA55C, even though VTPR data provide an additional constraint from late 1972. It perhaps should be recognised however that radiosonde observations continue to provide a constraint on the stratosphere from 1979 onwards. Satellite radiances (particularly from the TOVS instruments flown from 1978 until phased out between 1998 and 2006)have significant biases, and radiosondes play an important role in the biascorrection schemes for radiance data used by reanalysis centres, at least prior to the availability of substantial amounts of GPS radio occultation data from 2006 onwards. The betterquality reanalyses produced for the period from 1979 onwards is due to the combined use of radiosonde and satellite data, notwithstanding the labelling of the period as the satellite era.

*This paragraph has been reworked to provide a better overview of the study, and to better reflect the presence of satellite data products from prior to 1979 as well as the continued value of radiosonde data.*

(17) Page 12, line 15. be should be been.

*Corrected.*

(18) Page 12, line 21. Some rewording is required here, as it is a bit misleading to categorize the sudden warmings in fulldata reanalyses as a result of assimilated observations. They are a result of assimilating observations making use of a forecast model, and as such are a result of both forecastmodel dynamics and assimilated observations. The reanalyses that assimilate only surface observations demonstrate that assimilating upperair observations is important, but does not show that the forecast model is unimportant.

*I agree with your underlying point, but I don't think its fair to infer what you suggest from the text. The sentence is comparing the respective roles of the forecast model and the observations in constraining the event dates in surface input reanalyses, stating that the former is more important than the latter. Nowhere does it say anything about full-input reanalyses; if you insisted on inferring the converse it would be that assimilated observations are more important than the forecast model, not that the forecast model is unimportant. They could just as well be of equal importance in full-input reanalyses.*

(19) Page 12, lines 22 to 25. Newer reanalyses apply bias corrections to radiosonde data (generally following the work of Haimberger), and assimilating the biascorrected radiosonde data tends to control the biases of the reanalyses, at least in places and at levels radiosonde data are reasonably plentiful. No biascorrection of pre1979 radiosonde data was applied in ERA40, but the radiosonde data would nevertheless have limited systematic error in ERA40 to some extent. A change is not called for at this point in the paper, but consideration could be given to writing something earlier in the paper on this point.

*While I appreciate this comment as this is exactly the kind of impprovement that should bring improved confidence in the representation of 'radiosonde' era circulation, it is difficult for me to see how to explicitly tie this bias-correction to systematic errors in general (that, for instance, may not occur where observationns are directly available). It wasn't clear to me where to add this point to the text.*

(20) Page 12, lines 29 and 30. Comment (4) above, concerning the final sentence of the abstract, applies equally to this final sentence of section 6.

*I have mentioned ERA-5 and JRA-3Q here.*

**Response to Reviewer 2**

**General Comments**

The author presents a detailed comparative analysis of the quality of reanalyses data prior to 1979 and their potential inclusion in dynamical studies. In particular, he focuses on the analysis of relevant fields for the stratosphere-troposphere coupling. The results indicate that reanalysis data in the pre-satellite era is of sufficiently high quality to be considered together with the data of subsequent decades in these dynamical studies.

The manuscript is well written and the topic is certainly interesting for the scientific community, particularly that focused on stratosphere-troposphere coupling. The methodology applied for evaluating the quality of reanalysis data is also very thorough. However, in some cases I find the text a little bit dense particularly when describing Figures 6 and 7 and Section 4 and it would be advantageous for the manuscript to try to simplify that description. Thus, I recommend the publication of the manuscript after the mentioned minor correction and some other slight changes indicated below.

*In light of this and other reviewer comments I have spent some time trying to make this discussion clearer and more straightforward.*

**Specific Comments**

Page 5 Lines 5-8: The author indicates that the shift of the seasonal peak of SSWs in the satellite era with respect to the whole period (1958-2010) is only just due to the consideration of a longer database. I think the author could discuss a little bit more about this. Otherwise, the reader might get the impression that this is only a possible bias due to the lack of assimilated satellite data in the pre-satellite period. In contrast, it could be also related to multidecadal climate variability. Indeed, there are some studies that have also shown a change in the seasonality of SSWs in model simulations (e.g.: Ayarzagena et al. 2013). Finally, I would recommend citing here Gmez-Escolar et al. (2012) that already showed the change in the seasonal distribution of SSWs between the pre- and post-satellite periods.

*I have added a citation to Gmez-Escolar et al. as suggested. While it is possible that this reflects some true shift of the statistical seasonality of sudden warmings, it is also completely consistent with the null hypothesis that this is a result of sampling variability from a stratospheric climate that has not changed. This can be regarded as a source of decadal variabiltiy, but the statistics being what they are, it seems most reasonable to stick with this null hypothesis. This is also relevant to a point raised by Reviewer 3; I have added a bit of text in the discussion on this point as well.*

Page 7 lines 23-25: Maybe I am getting something wrong but the largest spreads, at least for the zonal wind, are found in the Northern Hemisphere.

*This is a good point - this is true of the upper stratosphere winds. I have been more careful to mention this in the text.*

Page 8 lines 11-12: Please indicate why you are selecting different levels for the stratospheric field in the Northern and Southern Hemisphere.

*This is reasonable; I had chosen a lower height to see if there was a level where the Southern Hemisphere might be better constrained, but having looked at this again it does not make a big difference and so I have changed the figure to show 30 hPa for both hemispheres to avoid having to explain any differences.*

Page 9 lines 20-25: I think the author should be careful with the description of the results in this paragraph. For instance, some fields that are indicated as not shown ( for T in JJA, for DJF u) are in fact shown and some others described as shown are not ( for T in DJF, for u in JJA). It would also help if a reference to the plots is included in each case too.

*This has been corrected and some additional explicit referencing of figure captions has been added.*

Page 10 lines 32-35: I might agree that data of 1950s may be of interest, but the results for NCEP/NCAR reanalysis for that decade are not shown in Figure 9.

*This paragraph has been removed.*

**Technical comments**

Page 4 Line 21: then  than

Page 4 Line 31: I think it would be better to write from 1958 to 2016.

Page 5, equation 2: in the second sum the upper limit should be N r instead of N s .

Page 5 equation 2: Please define N t

Page 7 line 15: I think it is the winter upper stratosphere.

Page 7 line 25: in many regions in  in many regions it

Page 8 line 25: Southern Hemisphere

Page 10 line 26: Please delete at.

Page 10 line 30: Please include ) after 9.

Page 11 line 6: reduced  reduce.

*These have all been corrected.*

**Response to Reviewer 3**

This is an interesting and well thoughout study about the potential value of earlier, pre-satellite era reanalysis records. It is important to quantify the potential value of this earlier period, as it is a major undertaking for a reanalysis center to provide pre- satellite reanalyses. With the exception of JRA-55 (and the ERA5 analysis, currently in production), most of the state-of-the-art full input reanalyses do not begin until 1979 (ERA-I, MERRA, CFSR) or 1980 (MERRA2).

I recommend publication of the manuscript pending consideration of the comments below. They are mostly minor, in that I leave them to the authors discretion, but I hope that responding to them would improve the impact of the paper. (An exception is that the author does need to better define a few things, to ensure the results are reproduceable. But this will be easy to do.) My more philosophical question about the proposed metric for assessing the value of earlier reanalyses (see below) is perhaps trickier to fully answer, and might be something for future work. I think that the contributions of this paper are already worthy of publication. Given that it could be a subject for future research, Ill sign this review, as I would welcome discussion with author.

Edwin Gerber

*Thank you for your comments.*

**General comments**

1) A few key elements of the procedure were not sufficiently documented. In particular, how were the SSW dates set, and how were the events classified in the spilits or displacements. I suspect this was done within the S-RIP Chapter 6 framework, assembled by Amy Butler. If so, I am not sure how to properly cite this information at this time, though they will be published. In any case, to reproduce these results, the reader does need to know the dates, and some insight on how they were obtained.

*This is the case; I have added some details to the text regarding the event definitions, but have not added an explicit list of dates. This could easily be done if deemed necessary.*

2) It would help the reader to adopt a consistent use of the nomenclature "full-input", "conventional-input", and "surface-input" throughout the paper. I appreciate that terms evolved in parallel to this research, but as a result of this time mismatch, they appear inconsistently through the text.

*These terms have now been explicitly defined (in reference to Fujiwara et al. 2017) and used more consistently throughout the text.*

3) I very much appreciate the central result of the manuscript: equation (3) and surrounding discussion, which seeks to quantify the value of earlier records. I was admittedly surprised, however, that the metric indicates that there is considerable value to much of the data in the austral hemisphere, where we know that the large scale circulation is not consistently captured by the reanalyses. (In Gerber and Martineau, 2018, for example, we found that the southern annular mode indices in JRA-55, ERA-40, and NCEP-R1 share only a small fraction of the variance during the pre-satellite period, indicating that there is very little consensus on the large scale state of the austral hemisphere on synoptic time scales.)

I think the key is the assumption that reanalyses properly capture the dynamical uncertainty, $\sigma_d$, in both the satellite and pre-satellite periods. I think this effectively implies that we trust their climatological values and variance, even if the become untethered to observations.

To make my concern clear, consider the extreme case where the reanalyses are perfect in the satellite era ($\alpha_s \to 0$) and know absolutely nothing about the state of the atmosphere in the radiosonde era ($\alpha_r \to \sqrt{2}$). In this case, $f \to 2$ and $\delta \to (1 - 2\beta) / (1 + (1 - \beta)2)$

When $\beta$ becomes small (¡0.5), you would still conclude that there is value in the reanalysis, even though it knows nothing about the state of the atmosphere. (The "real" $\beta$ is about 0.6, so in this limiting case delta would be negative, and you would concluded there is no value in earlier records). But given that $\alpha_s$ is not zero, and there is some limited skill in the radiosonde era, its not hard to see why delta is positive. And by this logic, there would be considerable value in using the entire record from ERA-20C, where beta drops below 0.5!

My intuition if we want an observationaly constrained estimate of the uncertainty, then we should only include the information from the earlier period when $\alpha_r < 1$. That is, when uncertainty in the renanalyses reaches the level of dynamical uncertainty, then we can argue the reanalyses are sufficiently untethered from the real atmosphere to provide any additional information than you could obtain from simply running a forecast model untethered to observations.

I havent thought this through enough to provide a good way to quantify the value of events when $\alpha_r < 1$. It helps me to think of this interms of events (as with the SSW composites shown by the author.) Suppose you have N events from the satellite record. Looking at past events, the idea would be to quantify the additional information content of each radiosonde period event on an event-by-event basis. When the spread between reanalyses for an earlier event is equivalent to the spread between events in the satellite period ( $\alpha_r = \alpha_s$), the event is clearly of complete value ($\delta = 1$); it should be added fully. Now your composite is based on N+1 events, and the uncertainty drops accordingly.

If the spread between the reanalyses for the event, however, becomes equivalent to the climatological/dynamical spread ($\alpha_r = 1$) then I feel that theres no additional information to be gained than if you simply ran a free running model: this event should be given zero value. I am just not sure how to develop a meaningful way to interpolate inbetween these cases.

*Perhaps the central issue here is that (3) measures the contribution towards reducing the variance of the sample mean from two samples drawn from a population with the same mean. In essence it is staring from the assumption that the forecast model has the same climatology as the real atmosphere, and if this really was the case then it would be worth including data from a period when the model was completely untethered from observations, since the longer time period would still act to reduce sampling uncertainty. So this metric does not tell us everything we need to know about how much information comes from the real atmosphere versus how much from the forecast model.*

*The sensible further criterion is whether the forecast model is actually following fluctuations that occured in the real world. In this case the spread between multiple reanalyses is being used to estimate $\alpha$ (in either case), and in the case where they all have the same variance as the real atmosphere but are fully independent realizations, $\sigma_o$ should approach $\sqrt{2}$ times the variance of the reanalyses. (This is laid out a bit more explicitly in the text now.)*

*So long as $\sigma_d$ in the reanalyses isn't too far off, $\alpha \to \sqrt{2}$ (in either era) is what one expects if the forecast model is just doing it's own thing. It's probably reasonable to set the bar rather lower than $\sqrt{2}$, but how far is a matter of judgement or of further criteria. The colouring adopted in Fig. 6 used 0.1, 0.3, and 1.0 as thresholds and so reflects the value of 1 you've suggested, but this is to some extent arbitrary.*

*As discussed further below, while this discussion emphasizes the difficulty of where to draw the line between including or not including the radiosonde era, a more important take away here is that so long as the $\alpha$'s are small relative to $\sigma_d$ (as is particularly the case for zonal wind in the NH stratosphere), it doesn't matter how much bigger $\alpha_r$ is relative to $\alpha_s$, it's still worth including the radiosonde period.*

*Another criteria one could think of is the possibility of systematic bias, but this kind of reanalysis intercomparison can't speak directly to that. The text gives a rough argument for when this can be neglected relative to $\sigma_d$. (The discussion of Fig. 10 is also relevant.)*

*From an event point of view, if the reanalysis is actually capturing the same event as the observations, $\delta$ is the relevant measure and serves to quantify your example.*

*In view of these points, I've significantly reworked the discussion of (3) to clarify these considerations, and I've added more emphasis to the importance of $\alpha_{r,s}$ being small.*

4) I appreciate that the comment above is weak on specific suggestions. To be more concrete, I would have appreciated more discussion of the different limits around equation (3). The limit where $\alpha_s$ is small and $\alpha_r$ approaches sqrt(2) was interesting to me, as it drove home this issue of whether we ought to trust a good model that is untethered to reality.

Another problematic limit is $\alpha_r = \alpha_s$. Here, you always use more data, even if its all untethered to reality. (Based on my arguements above, the value of the reanalyses should be zero when $\alpha_r$ or $\alpha_s$ approaches 1.)

*The problem here is really just the same as above; when either $\alpha_r$ or $\alpha_s$ approaches $\sqrt{2}$ this is indicative of the reanalyses becoming untethered to the observations in the respective period. If they are both roughly equal, the two eras contribute equally to reducing the uncertainty as should be the case.*

And not to be overly critical, Figure 4 was not easy to interpret. Consider using color or more simply marking the contours. (I know that I should have realized that diagonal is 1 by definition, but it took me time at first reading.) I also think that its inappopriate to show such a range. Once $\alpha_s$ or $\alpha_r$ reach $\sqrt{2}$, nothing is tethered to observations, and I dont see how $\delta$ is meaningful for values beyond this point.

*This is a good point, the range shown has been reduced accordingly. I have also added labels to the contours.*

5) There is a paper that can be cited for the Martineau data set: Martineau, P., Wright, J. S., Zhu, N., and Fujiwara, M.: Zonal-mean data set of global atmospheric reanalyses on pressure levels, Earth Syst. Sci. Data, 10, 19251941, doi:10.5194/essd-10-1925-2018, 2018b.

*This has been added.*

Also, its my understanding that MERRA2 has a DOI that should be cited, as its very important for them to justify resources. I find the situation problematic, in that you got the data from a different source (which did cite this doi), but perhaps you could add the doi to the data section.

*As you say, given that the dataset I've used is not the reanalysis center itself, it seems more appropriate to cite the Martineau paper/dataset, especially since there is a citation for each reanalysis already.*

**Small comments** 2:3-8 This would be a good time to differentiate and define full-, conventional- and surface-input reanalyses.

*I've done so a bit later, when discussing the reanalysis datasets in Section 2.*

2:21

citep[e.g.,][] (Also, I think that perhaps one should include the comma on e.g., since if you spelled out the phrase, it would be: for example, Matsuno 1971. But perhaps this is a case where American English is different from British.)

*Done.*

2:34 I might break this off as a full sentence, instead of using the semicolon.

*Done.*

3:11 I appreciate why the author states that they are constrained *primarily* by surface observations (as the reanalyses are given changes in radiative gases, etc.), but this sentence seemed a bit to vague.

*I have reworded this to emphasize that upper-air observations are not assimilated by these products.*

3:25 consider rephrasing this sentence: I understood it completely, but had to re-read it a few times

*I have reworded, hopefully it is clearer.*

4:16 This would be a place to explain how the dates were set, and how splits and displacements were classified, or at least point the reader to the necessary information.

*This has been done.*

4:24-26 This is a fascinating/perplexing result. I think it makes sense, though: ERA- 20C does a good job of getting SSWs, but since it only gets the dates right for half of them, you are better off treating it as a free running model (i.e. not fixing dates to reality) than trying to make it conform with what actually happened in our atmosphere. ERA-20C provides the challenge to your metric in equation 3: I do think you would argue that its worth while using the entire record, even if just assume it knows nothing about the actual state of the atmosphere. Thats what motivated my thoughs on comment 3 above.

*If I've understood you correctly, this is a case where (3) would not see a difference between the satellite era and the radiosonde era, and would therefore weight the whole record evenly. As you suggest, this is a case where the second criterion is important; for the cases shown in Figs. 6, 8, and 9, α is generally smaller than expected if ERA-20C was completely untethered to observations, but still or the order 1.*

4:29 splits and displacements need to be defined

*The method of Lehtonen and Karpechko has been used; this is now stated explicitly. The reduction of CI's is the same for other definitions that I've tried.*

5:1 second half of the line is awkwardly phrased

*Reworded.*

5:17 Im concerned that the zonal mean wind at 60 N and 10 hPa is decidely not Gaussian, and rather skewed towards negative values.

*This is true, and so for climatological averages this could be an issue - but as is stated in the next sentence, the central limit theorem helps out. It's also less clear this is as important for things like composites which may give rise to more gaussian statistics. More generally, getting into a detailed discussion of the corrections one might have to adopt in the presence of non-gaussianity seems a distraction at this point.*

5:18. There is a sentence between when you introduce $\sigma_d$ and $\sigma_o$ and define them. Consider moving the first sentence of the next paragraph up, to define the variables, before discussing the central limit theorem.

*This would move the comment about non-gaussianity even further from the example given for X; but I've slightly changed the discussion to bring the definition closer to the introduction of these terms.*

6:10 In the limit where the reanalysis error is small relative to the dynamical uncertainty, isnt f small, and delta about equal to 1?

*To $O(\alpha_{r,s}^0)$, yes. To leading order in $\alpha_{r,s}$ the expression is as given. β enters at the next order of the expansion (as $1 - \beta$, in fact), and this term is less than a 10% correction to δ for δ as small as 0.5, so long as beta isn't too small. See also the response to a later comment.*

8:20-24 This sentence is long. Consider breaking at the ;, and then being more clear what agreement you mean to refer to.

*Reworked as suggested.*

Fig. 8: I assume the Fourier analysis was done on the deaseasonalized winds, as theres no discernable annual cycle peak here!

*Yes, this is the case. This is now stated in the text.*

10:21 and Figure 7 e,f I am confused how you can estimate δ without knowing β. It only seems to decouple from β when $f$ is small. And in this limit, δ would be close to 1 (and its sort of a trivial result: you trust everything.) In the figure, the value of delta varies considerably (changing sign!) so you must have some finite value of β. What is 0.6?

*The calculation was done assuming a value of 0.6 for β; in light of the discussion above around the importance of α itself, I've changed this figure to show α estimated on a month-by-month basis. This also removes the question of*

*the dependence on $\beta$.*

11:1-2 This sentence could be split up, giving you two sentences, enough to justify a paragraph!

*Done.*

11:13-15 It was hard for me to see this important result. At some latitudes (c. 65 in panel a of Fig 10, or near 75 in panel b), the uncertainty bounds on the full record were larger than than for the satelite record. So clearly there wasnt always a 20% reduction. Im was also rather struck by the fact that the dashed curved in panel b of Figure 10 approaches the edge of the confidence interval on the "all" composite. Does this mean that they were almost statistically different, or would this only apply when the confidence intervals themselves separate.

*The confidence intervals are themselves statistical estimates which will only approach the 20% reduction in a probabilistic sense; so yes, while there are regions where this does not hold, it is broadly the case that there is a reduction of this order. I have made it clearer that this reduction is not universal. As for the differences, perhaps you meant the blue dashed curve in panel (c)? In any case both are in fact within the confidence intervals given. Although I have not carried out an explicit test of statistical differences this seems unlikely.*

From a practical standpoint, if I wanted to ask whether my model was signficantly different from our best estimate of observations, which error bound should I use?

*On the weight of evidence presented (and reviewed) here, the bounds based on all data available. This has been emphasized in the text.*

11:30 Might be good to emphasize "has been quantified in equation (3)."

*Done.*

12:2 Related to some comments above, when the dynamical uncertainty dominates, doesnt this imply that you trust everything?

*Yes - as discussed above this is a key message. (Although a better way to think of it might be that everything is valuable to reducing this uncertainty). I have re-emphasized it in the text.*

12:15 An opportunity to use surface-input nomenclature.

*Taken.*

15:18 I am not sure how to see this in Figure 9. Doesnt the fact that $\delta$ is consis- tently greater than 0 for ERA-20C imply that theres always value to be found from this reanalysis?

*I have rewritten this paragraph emphasizing points that have already come up in the discussion above.*

15:26-30 Could be opportunity to highlight that your message has been heard, and ERA5 hopes to go back to 1950.

*Done, also following suggestion of reviewer 1.*

Fig 4: See my general comment (4) above. I think this figure could be improved.

*I have reduced the range shown and added labels to the contours.*

Fig 5: Your notation differs a bit here, $\sigma_d$ vs. $\sigma_{dyn}$. Its clear enough for the reader, but consistency is best.

*Fixed.*

Fig 7: caption has the wrong symbol. I would have appreciated more detail here in how the bottom panels were computed.

*The caption has been corrected.*

Fig. 8: I find that the log scale makes comparison very difficult. Would it be possible to show the ratio of the differences in the power spectra? This is a number that would presumable vary from 0 to about 2 for all timescales. (It would be 2 if the limit that the models become untethered from observations. I guess it could become larger if there are systematic biases.)

*I've taken the suggestion to show ratios (and added a line at 2 for reference). I do think this makes the figure and its interpretation clearer.*

If nothing else, the reference time series of JRA-55 gets buried by the other lines: consider bringing it up to the top. (If you produced this plot with matlab, but want to keep it first in the legend, a solution is to just print it again.)

*Following your first suggestion, JRA-55 is no longer shown.*

Figure 10: Are these 95% confidence intervals?
*Yes, this has now been clarified in the text.*